# Meteorins regulate the formation of the left-right organizer and the establishment of vertebrate body asymmetry

**Fanny Eggeler[1], Jonathan Boulanger-Weill[1†], Flavia De Santis[2,3†], Laura Belleri[1†], Karine Duroure[1], Thomas O Auer[2,4], Shahad Albadri[1]\*, Filippo Del Bene[1]\***

[1]Sorbonne Université, INSERM, CNRS, Institut de la Vision, Paris, France; [2]Institut Curie, PSL Research University, Paris, France; [3]ZeClinics SL. Sant Feliu de Llobregat, Barcelona, Spain; [4]Department of Biology, University of Fribourg, Fribourg, Switzerland

**\*For correspondence:**
shahad.albadri@inserm.fr (SA);
filippo.del-bene@inserm.fr (FDB)

[†]These authors contributed equally to this work

## eLife Assessment

This study presents **important** insights into the regulation of left-right organ formation. By combining genetic perturbation of all three Meteorin genes in zebrafish and timelapse imaging, the authors identify an essential role for this protein family in the establishment of left-right patterning. They provide **convincing** evidence that Meteorins are required for the morphogenesis of dorsal forerunner cells, the precursors of the left-right organizer (also named Kupffer's vesicle) in zebrafish. In line with this, Meteorins were shown to genetically interact with integrins ItgaV and Itgb1b to regulate dorsal forerunner cell clustering.

**Abstract** While the exterior of vertebrate bodies appears bilaterally symmetrical, internal organ positioning and morphology frequently exhibit left-right (L-R) asymmetries. In several vertebrates, including human, mouse, frog, and zebrafish, left-right symmetry-breaking during embryonic development is initiated by a ciliated organ called the Node or left-right organizer. Within the Node, a leftward flow of extraembryonic fluid named the Nodal flow mediates the asymmetric expressions of Nodal factors. Although downstream Nodal pathway components leading to the establishment of the embryonic left-right axis are well known, less is known about the development and formation of the embryonic Node itself. Here, we reveal a novel role for the Meteorin protein family in the establishment of the left-right axis and in the formation of Kupffer's vesicle, the Node equivalent structure in zebrafish. We show that the genetic inactivation of each or all three members of the zebrafish Meteorin family (*metrn*, *metrn-like a*, and *metrn-like b*) leads to defects in properties of the Kupffer's vesicle, caused by impaired assembly and migration of the Kupffer's vesicle forming dorsal forerunner cells. In addition, we demonstrate that Meteorins genetically interact with integrins ItgαV and Itgβ1b, regulating the dorsal forerunner cell clustering, and that *meteorins* loss-of-function results in disturbed Nodal factor expression and consequently in randomized or symmetric heart looping and jogging. These results identify a new role for the Meteorin protein family in the left-right asymmetry patterning during embryonic vertebrate development.

## Introduction

From the outside, the vertebrate body plan appears bilaterally symmetric. However, internal organs positioning and morphology often display left-right (L-R) asymmetries. For instance, in vertebrates,

the heart generally lies on the left side, the liver and the pancreas are positioned on the right, and the gut presents asymmetric rotations.

One major regulator for the establishment of the L-R axis is Nodal, a ligand belonging to the TGFβ protein family (*Blum et al., 2014*; *Blum and Ott, 2018*; *Grimes and Burdine, 2017*; *Kawasumi et al., 2011*; *Branford and Yost, 2004*; *Zhou et al., 1993*). Its signaling is activated in the left lateral plate mesoderm (LPM), whereas Nodal remains inactive in the right LPM, creating an embryo-scale left-right asymmetry (*Grimes and Burdine, 2017*). In mice, this asymmetry is achieved by an oriented rotation of the cilia of a structure called the Node, generating a leftward flow of extraembryonic fluid named the Nodal flow. This results in an asymmetric expression of several Nodal factors and Nodal signaling. Participating in this patterning, Leftys, soluble inhibitors belonging to a subclass of TGF factors, antagonize Nodal signaling (*Chen and Schier, 2002*; *Chen and Shen, 2004*).

The Node and the Nodal flow have been described in numerous vertebrates. For instance, in *Xenopus* the gastrocoel roof plate and in zebrafish the Kupffer's vesicle (KV; *Essner et al., 2005*; *Vick et al., 2009*; *Nonaka et al., 2002*; *Schweickert et al., 2007*; *Okada et al., 2005*; *Tanaka et al., 2005*) are key structures of the L-R symmetry breaking during the embryonic development (*Nonaka et al., 2002*; *Okada et al., 2005*; *Nonaka et al., 1998*). Similarly, in humans and other mammals like rabbits and mice, the systematic establishment of left-right body symmetry begins with asymmetric fluid flow driven by rotating cilia (*Smith et al., 2019*). This occurs within the transient primitive node in humans or the embryonic node in mice and rabbits (*Okada et al., 2005*; *Nonaka et al., 1998*; *Schoenwolf et al., 2014*). However, in other mammalian classes, although left-right symmetry breaking mechanisms are believed to be cilia-independent, they are still poorly understood at present (*Blum and Ott, 2018*; *Hamada et al., 2002*). In chicks, the symmetry-breaking structure known as Hensen's node serves as the ciliated organizer responsible for the left-right asymmetry establishment. In this species, the process of symmetry breaking is believed to occur independently of the non-motile cilia present within Hensen's node. Instead, the breaking of symmetry is primarily attributed to the leftward movement of cells around the node (*Gros et al., 2009*; *Essner et al., 2002*; *Stephen et al., 2014*).

Compared to the relatively flat-shaped mouse node, the zebrafish KV is a fluid-filled sphere with a ciliated epithelium, and it is formed by dorsal forerunner cells (DFCs; *Essner et al., 2005*). DFCs first emerge as cells of the epithelium of the dorsal surface in direct contact with the yolk syncytial layer at 6 hours post-fertilization (hpf; *Oteíza et al., 2008*). Around 8–9 hpf, following the epibolic movement, 20–30 DFCs form a single cluster and migrate towards the vegetal pole in a non-involuting manner at the leading edge (*Oteíza et al., 2008*; *Cooper and D'Amico, 1996*). During this migration process, polarized DFCs organize into multiple focal points by retaining long-lasting apical contacts within the cluster, and remaining delaminated DFCs maintain contact with the cluster by cell-cell contact mechanisms (*Oteíza et al., 2008*; *Pulgar et al., 2021*). At the final phase of epiboly and after reaching the vegetal pole, the DFC cluster is separating from the dorsal marginal enveloping layer cells by losing the apical contacts (*Oteíza et al., 2008*; *Pulgar et al., 2021*). The focal points of the cluster are then rearranged by integrating the unpolarized DFCs to form a single focal point that will expand into a monolayer rosette structure containing a lumen to form the KV by 12 hpf (*Essner et al., 2005*; *Oteíza et al., 2008*; *Cooper and D'Amico, 1996*; *Pulgar et al., 2021*). Along this process, non-motile and motile monocilia are generated on the apical membrane facing the lumen that create a counterclockwise flow of fluid by the KV, called Nodal flow (*Oteíza et al., 2008*; *Kramer-Zucker et al., 2005*). It was shown that the leftward-directed flow within the KV results in asymmetric expression of zebrafish Nodal genes, like *spaw*, *lefty1*, and *lefty2*, which is fundamental for proper L-R patterning (*Hirokawa et al., 2006*; *Harvey, 1998*). However, the molecular mechanisms involved in correct DFC migration and clustering that will form the KV remain largely elusive.

In this context, it has been reported that *nodal, lefty,* as well as *p-smad2* expression is downregulated in *Meteorin*-null ES cells. As such, Meteorin (Metrn) was hypothesized to be a novel important regulator of Nodal transcription (*Kim et al., 2014*).

Meteorin (Metrn), a secreted neurotrophic factor highly conserved among vertebrates, is expressed during early mouse development. It is already detected at the blastocyst stage in the inner cell mass and then expands through the extraembryonic ectoderm to the central (CNS) and peripheral nervous system (PNS) during later developmental stages (*Kim et al., 2014*). It was first described to induce glial cell differentiation and to promote axonal extension in dorsal root ganglion (DRG) explants in vitro (*Nishino et al., 2004*). The disruption of *Metrn* function in mice resulted in early embryonic lethality

(*Kim et al., 2014*), preventing the investigation of Metrn protein function during early embryonic development in vivo and, in particular, their role in L-R patterning. Also conserved among vertebrates, its paralog Meteorin-like (Metrnl) has been first reported as a downstream target of the Pax2/5/8 signaling pathway during otic vesicle development (*Ramialison et al., 2008*) and was later shown to have neurotrophic properties comparable to the ones of Metrn (*Jørgensen et al., 2012*).

Here, using the zebrafish larva as a model system, we generated knockout lines for all three existing zebrafish *metrn* genes (*metrn*, *metrnla,* and *metrnlb*). While all three single and compound mutants are viable, *metrns* mutant embryos displayed organ patterning defects, notably with randomized or symmetric heart looping at a significantly higher frequency. Like in mice, *metrn* genes are expressed during early development already from the two-cell stage, and transcripts are detected in DFCs as well as in the KV structure. We show that *metrns* loss of function leads to DFC disorganization and Nodal flow formation defects within the KV. Together, our study reveals a critical role for Metrn proteins in the DFC clustering, KV formation, and in the establishment of the L-R axis.

## Results

### Metrns loss-of-function affects proper heart looping and correct visceral organs positioning

To study the functions of Meteorin proteins during early development, we generated knockout zebrafish lines using the CRISPR/Cas9 technology to target the second exon of each gene that encodes for the signal peptide sequence. The newly generated zebrafish mutant lines carried out-of-frame deletions in the coding sequence of *metrn*, *metrnla,* and *metrnlb* (*Figure 1—figure supplement 1A*). In contrast to published embryonic lethal *Metrn* mouse mutants (*Kim et al., 2014*), constitutive homozygote zebrafish mutants for all three genes individually (*metrn*[-/-]; *metrnla*[-/-]; *metrnlb*[-/-]), in double (*metrn*[-/-], *metrnla*[-/-]; *metrn*[-/-], *metrnlb*[-/-]; *metrnla*[-/-], *metrnlb*[-/-]) and triple (*metrn*[-/-], *metrnla*[-/-], *metrnlb*[-/-] or triplMut), were all viable and did not show any obvious morphological defects (*Figure 1A*).

We first validated that our mutagenic approach resulted in the generation of null alleles. To do so, we performed in situ hybridization on *metrn*, *metrnla,* and *metrnlb* from two-cell stage to 1 day post-fertilization (dpf) on triplMut embryos (*Figure 1—figure supplement 1B*). No expression could be detected for neither of those two genes in the triplMut embryos, indicating the degradation of the transcripts in the *metrn* mutant backgrounds (*Figure 1—figure supplement 1B*). To confirm this observation, we compared the expression level of all three *metrn* genes upon total mRNA extraction from 14 and 48 hpf WT and triplMut embryos by qRT-PCR. *Metrn* and *metrnla* expression was detected in samples of 14 hpf WT embryos, and *metrnlb* expression was detected in samples of 48 hpf WT embryos. A significant reduction in the expression of all *metrn* genes could be measured using triplMut samples of the same age. These results suggest that the mRNA produced by the loci that we targeted is degraded by nonsense-mediated decay (*Figure 1—figure supplement 1C*).

Despite the absence of gross morphological defects, using *myl7* antisense riboprobe in situ hybridization in single mutant embryos, we observed a mispositioning of the heart in triplMut mutant embryos. Indeed, while 88.43% of the wild type control embryos had D-looped shaped hearts, only 38.17% of the triplMut embryos exhibited a WT-like heart looping phenotype (*Table 1* and *Figure 1B–C*). Similarly, *metrn*[-/-] and *metrnla*[-/-] single mutant embryos displayed a high proportion of randomized heart phenotypes, whereas *metrnlb*[-/-] single mutant embryos showed a small percentage of heart looping phenotypes (*Table 1* and *Figure 1B–C*). The injection of *metrn* and/or *metrnla* mRNA into one-cell stage triplMut zebrafish embryos could partially rescue the observed phenotypes (*Table 1*, *Figure 1—figure supplement 1D*). In addition to the heart, morphological L–R asymmetry of the visceral organs was also perturbed in triplMut embryos (using *gata6* expression as a marker). Indeed, 10.49% of the triplMut embryos exhibited symmetrical placement of the gut and pancreas with respect to the midline and 9.88% *situs inversus* or heterotaxy phenotypes compared to wild type embryos (WT: 93.84% L–R asymmetry, 4.79% L–R symmetry and 1.37% heterotaxia; *Figure 1D–E*). These results indicate that while Metrns are not implicated in the specification of neither the heart nor the visceral organs (*myl7* and *gata6* expression being detected in triplMut embryos and in single mutants for *metrns*), their loss of function reveals that they are implicated in the L–R asymmetry positioning of these organs. The left-jogged and D-looped heart or the right-positioned liver are two of many examples of early L-R asymmetry determination in zebrafish and are well-studied readouts for altered L-R

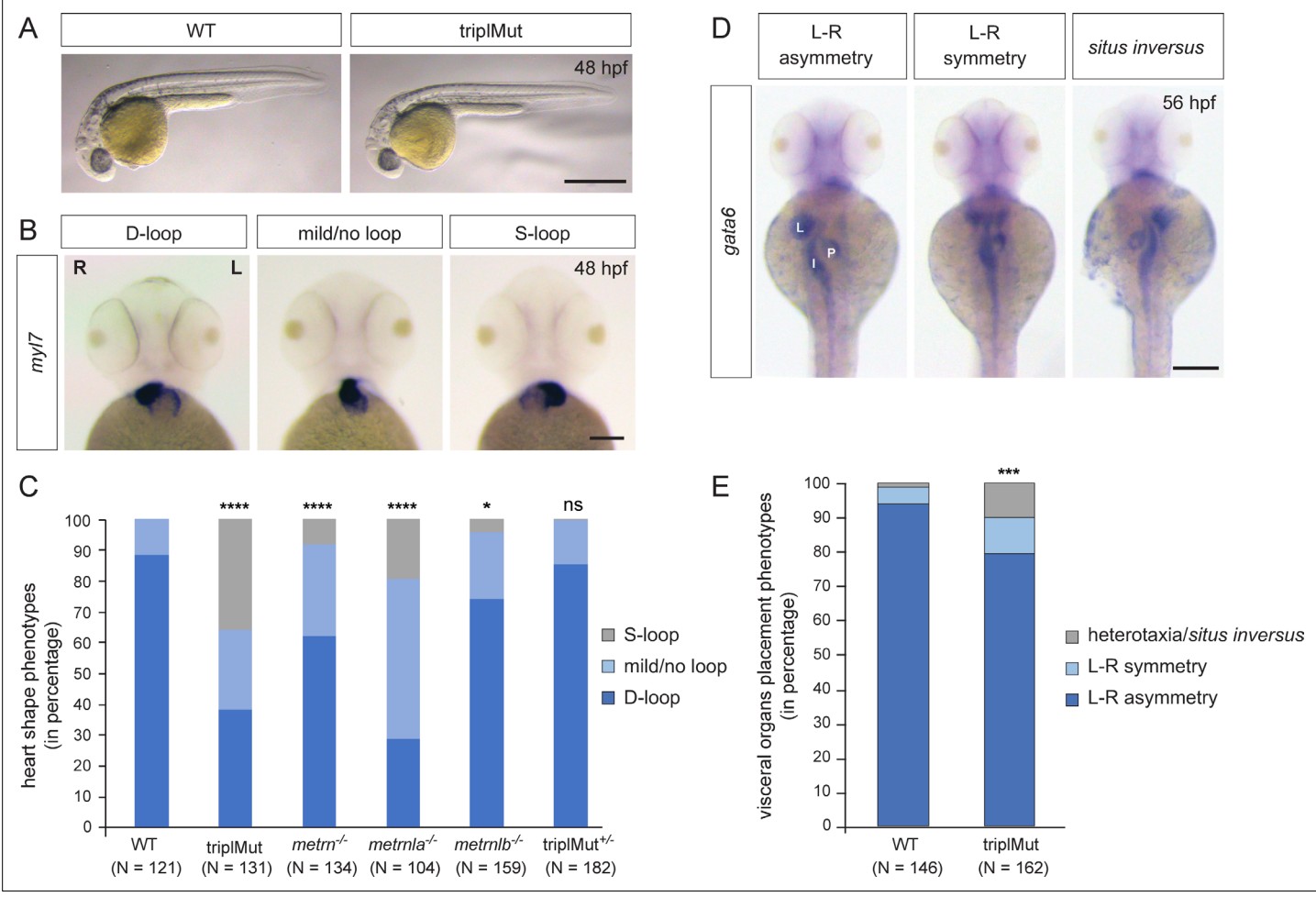

**Figure 1.** Metrns loss-of-function causes heart jogging/looping and visceral organ positioning defects. (**A**) 48 hpf triplMut zebrafish display no gross phenotypic defects compared to wild type (WT) embryos. (**B**) Examples of 48 hpf embryos showing mRNA expression of the heart marker *myl7* and the different heart looping phenotypes (ventral view). (**C**) Quantification in percentage of heart looping phenotypes at 48 hpf in WT, triplMut, *metrn⁻/⁻*, *metrnla⁻/⁻*, *metrnlb⁻/⁻*, triplMut⁺/⁻ embryos; displayed p-values compared to WT: ****p-value:<1.0e-5 for triplMut, ****p-value:<1.0e-5 for *metrn⁻/⁻*, ****p-value:<1.0e-5 for *metrnla⁻/⁻*, *p-value: 0.003 for *metrnlb⁻/⁻*, ns p-value: 0.49 for triplMut⁺/⁻, p-value compared to *metrnlb⁻/⁻*: *p-value: 0.034 for triplMut⁺/⁻ (not displayed). (**D**) Example of 56 hpf embryos showing *gata6* mRNA expression highlighting the different visceral organ positioning phenotypes (ventral view). (**E**) Quantification in percentage of visceral organ positioning phenotypes at 56 hpf in WT and triplMut embryos, ***p-value: 0.00034. In (**B**) L=left, R=right. In (**D**) L=liver, P=pancreas, I=intestine. Scale bars in (**A**): 3mm, (**B**): 100 µm, (**D**): 500 µm.

The online version of this article includes the following source data and figure supplement(s) for figure 1:

**Figure supplement 1.** (**A–C**) *Metrn* genes CRISPR/Cas9 mediated knockout generation and validation. (**D**) *Metrns* loss-of-function effect on heart morphology.

**Figure supplement 1—source data 1.** qRT-PCR values for metrn, metrnla and metrnlb expression levels at 14 hpf in wt and triplMut backgrounds.

patterning processes. Therefore, our results suggest that Metrns, in particular Metrn and Metrnla, may have a crucial function in establishing L-R asymmetry during the early stages of development. To exclude potential genetic compensation effects among paralogs that were reported in other studies (*El-Brolosy et al., 2019*), we focused all subsequent analyses (if not stated otherwise) on the triple mutant line (triplMut).

## Metrns are required for proper Nodal factor gene expression

Nodal signaling is one of the major regulatory pathways for the determination of the L-R axis during vertebrate development. In zebrafish, null mutants for several Nodal factors like *spaw* (*Nodal* zebrafish orthologue), *dand5* or *lefty1* (*lft1*) all display symmetric or randomized heart looping and jogging similarly to *Nodal* mutant mice (*Montague et al., 2018*; *Long et al., 2003*; *Brennan et al., 2002*; *Saijoh*

**Table 1.** Quantification of heart looping phenotypes at 2 dpf using *myl7* riboprobe in situ hybridization on wild-type (WT), *metrn*s single and triple mutants (triplMut), and *metrn* and/or *metrnla* mRNAs-injected embryos.
'Mild/no loop' designates hearts with no particular L-R orientation. D-loop, dextral loop; mild/no-loop and S-loop, sinistral loop.

| | D-loop | mild/no-loop | S-loop |
|---|---|---|---|
| WT (N=121) | 88.43% | 11.57% | 0% |
| *metrn*[-/-] (N=131) | 61.19% | 29.10% | 8.21% |
| *metrnla*[-/-] (N=104) | 28.85% | 51.92% | 19.23% |
| *metrnlb*[-/-] (N=104) | 74.21% | 21.38% | 4.41% |
| triplMut (N=134) | 38.17% | 25.95% | 35.88% |
| triplMut[+/-] (N=182) | 85.16% | 14.29% | 0.55% |
| triplMut +*metrnla* mRNA (N=160) | 37.5% | 53.13 | 9.37% |
| triplMut +*metrn* mRNA (N=156) | 45.51% | 42.31% | 12.18% |
| triplMut +*metrnla* + *metrn* mRNA (N=162) | 52.47% | 27.16% | 20.37% |

*et al., 2003*). We therefore asked whether Metrns loss of function could affect the expression of these factors. To do so, we assessed *dand5*, *spaw*, *lft1*, and *lft2* expression in our *metrns* mutant embryos. By qRT-PCR, we measured a significant reduction of *dand5* expression at 14 hpf in triplMut embryos compared to wild type controls (*Figure 2A*). Similarly, the expression of *spaw*, *lft1*, as well as *lft2*, in the absence of Metrns proteins was also downregulated in triplMut embryos compared to wild type embryos (*Figure 2A*).

In order to assess whether Metrns loss of function altered the expression patterns of *spaw*, *lft1*, and *lft2*, we performed in situ hybridization experiments for all three genes at 16 hpf (*Figure 2B*). At 16 hpf, *spaw* is normally expressed adjacent to the KV and in the left LPM. In *metrn*[-/-] and *metrnla*[-/-] single mutants as well as triplMut embryos, we could not detect any *spaw* expression in the left LPM and only faintly around the KV (*Figure 2B*). The spaw inhibitor *lft1* at this stage is expressed along the midline. In all analyzed *metrn* mutants (single *metrn* and *metrnla* and triplMut), its expression was severely disrupted at the midline (*Figure 2B*, middle panel). The same was observed for *lft2*, which expression in the left heart field was either absent or randomized in *metrn* mutants, single or triplMut embryos, compared to the wild type embryos (*Figure 2B*, lower panel).

We next characterized *dand5*-specific R>L bias expression pattern in triplMut embryos by in situ hybridization. In more than 61% triplMut embryos, we could detect particularly reduced *dand5* expression at 14 hpf, and 23% displayed an abnormal *dand5* R>L expression bias (*Figure 2C–D*). In single *metrn*[-/-] and *metrnla*[-/-] mutant embryos, *dand5* expression pattern was also altered (*Figure 2—figure supplement 1*). These results are in line with previous findings reporting that *dand5* expression in the KV is only altered if upstream symmetry-breaking mechanisms are disrupted (*Pelliccia et al., 2017*; *Hojo et al., 2007*; *Gourronc et al., 2007*). These results indicate that Metrns are required for the proper expression of Nodal factor genes that are determinants of proper L-R patterning.

### *Metrns* are expressed during early zebrafish development

To investigate how Metrns affect the expression of Nodal factors during early development, we analyzed the pattern of expression of all three zebrafish *metrn* genes from two-cell stage to 48 hpf. We first investigated the expression pattern of *metrn*, *metrnla*, and *metrnlb* by in situ hybridization and hybridization chain reaction (HCR) experiments. At the two-cell stage, we detected both *metrn* and *metrnla* maternally expressed transcripts, whereas *metrnlb* expression was not detected (*Figure 3A*, *Figure 3—figure supplement 1A*). During gastrulation, the expression of *metrn* and *metrnla* could be detected, whereas *metrnlb* expression was again not observed in our in situ hybridization analysis. The earliest time point for zygotic expression of *metrn* and *metrnla* was at 6 hpf (shield stage). At this stage, *metrnla* expression could be detected over the whole animal cap, whereas zygotic *metrn* expression onset was observed only around the yolk syncytial layer (YSL) at this developmental stage (*Figure 3A*). *Metrn* expression was then restricted to the developing LPM and to the leading edge of the former shield area from 9 hpf (*Figure 3A*, *Figure 3—figure supplement 1B*). In comparison,

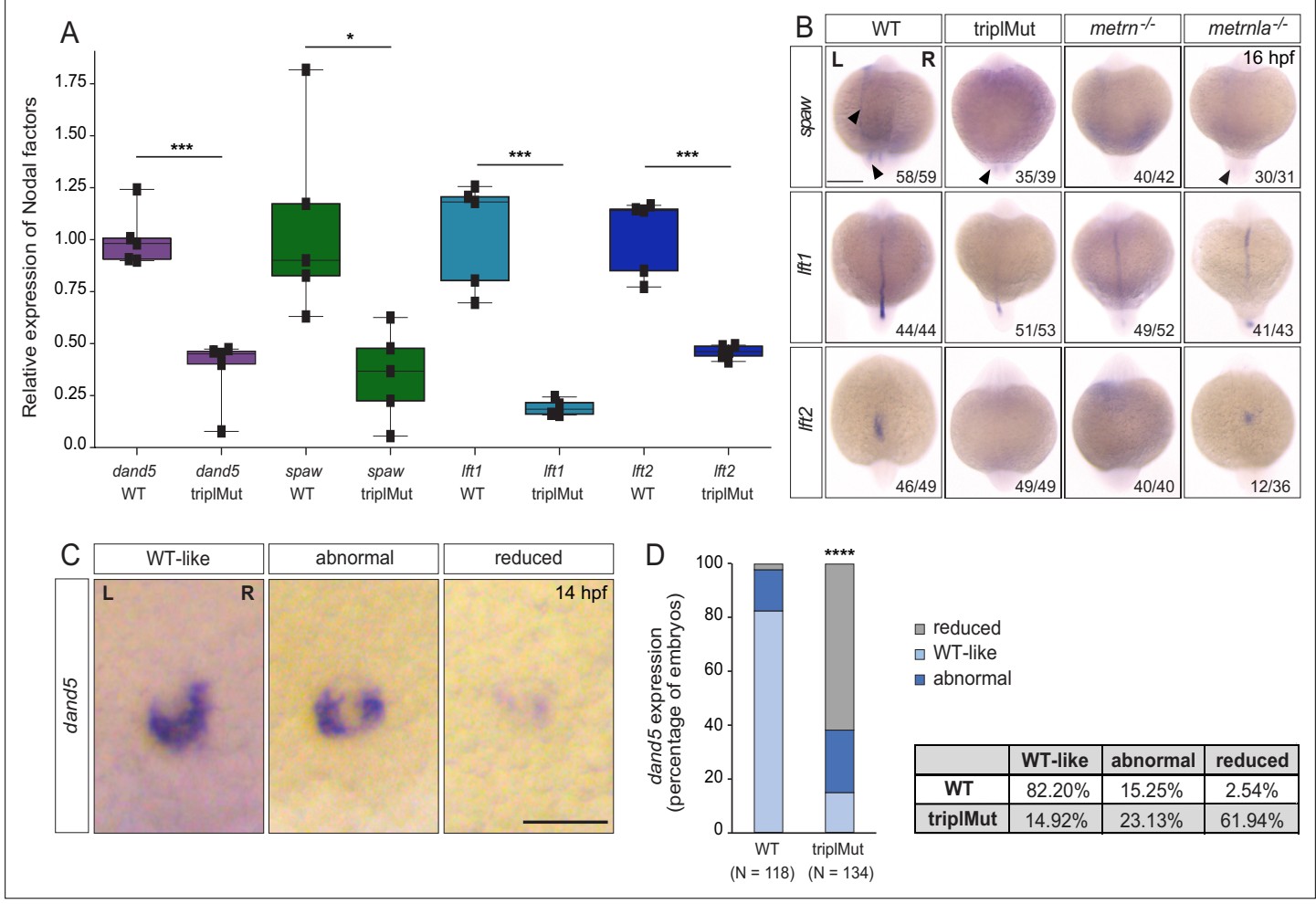

**Figure 2.** Metrns are required for proper Nodal factor gene expression. (**A**) qRT-PCR analysis for *dand5, spaw, lft1,* and *lft2* expression in 14 hpf wild type (WT) and triplMut embryos (Student t-tests, ***p-value: 0.00019 for *dand5;* *p-value: 0.014 for *spaw;* ***p-value: 0.0004 for *lft1;* ***p-value: 0.00019 for *lft2*). Error bars indicate standard deviation. (**B**) Dorsal view of 16 hpf triplMut, *metrn*[-/-], and *metrnla*[-/-] embryos with reduced expression of *spaw, lft1,* and *lft2* as revealed by in situ hybridization. (**C**) *Dand5* expression at 14 hpf as revealed by in situ hybridization showing different transcripts distribution of *dand5* mRNA. (**D**) Quantification in percentage of *dand5* expression phenotypes at 14 hpf in WT and triplMut embryos, ****p-value:<1.0e-5. Scale bar in (**B**): 250 μm, in (**C**): 100 μm; L=left, R=right. Source data for Figure 2A provided in '*Figure 6—figure supplement 2—source data 1*'.

The online version of this article includes the following source data and figure supplement(s) for figure 2:

**Source data 1.** qRT-PCR values for dand5, spaw, lft1, lft2 and spaw expression levels at 14 hpf in wt and triplMut backgrounds.

**Figure supplement 1.** Nodal factors gene expression is altered in the absence of Metrns.

*metrnla* was found expressed throughout the whole enveloping layer and developing midline (*Figure 3A*, *Figure 3—figure supplement 1B*). At 12 hpf, the expression of *metrn* was restricted to the KV area while *metrnla* transcripts were mainly found in the midline (*Figure 3A–B*). By 14 hpf, *metrn* expression was present in the KV area as well as in the developing brain, whereas *metrnla* transcripts were mainly found in the midline and the KV area (*Figure 3—figure supplement 1B*). In contrast, from the two-cell stage to 14 hpf, no *metrnlb* transcripts could be detected (*Figure 3A*, *Figure 3—figure supplement 1A-B*). By HCR, we found *metrn* and *metrnla* expression colocalizing in the area of the KV at 12 and 14 hpf (*Figure 3B*). By 24 and 48 hpf, both genes were found strongly expressed in the CNS (*Figure 3—figure supplement 1C*). Finally, *metrnlb* expression onset started from 24 hpf broadly in the embryonic brain (*Figure 3—figure supplement 1C*). At 48 hpf, *metrnlb* transcripts were mostly found in the otic vesicles and at the somite boundaries (*Figure 3—figure supplement 1C*).

We next analyzed the relative expression of *metrn* genes to landmarks of the KV structure, like *sox32*, a well-studied DFC marker (*Alexander et al., 1999*; *Kikuchi et al., 2001*). We observed a

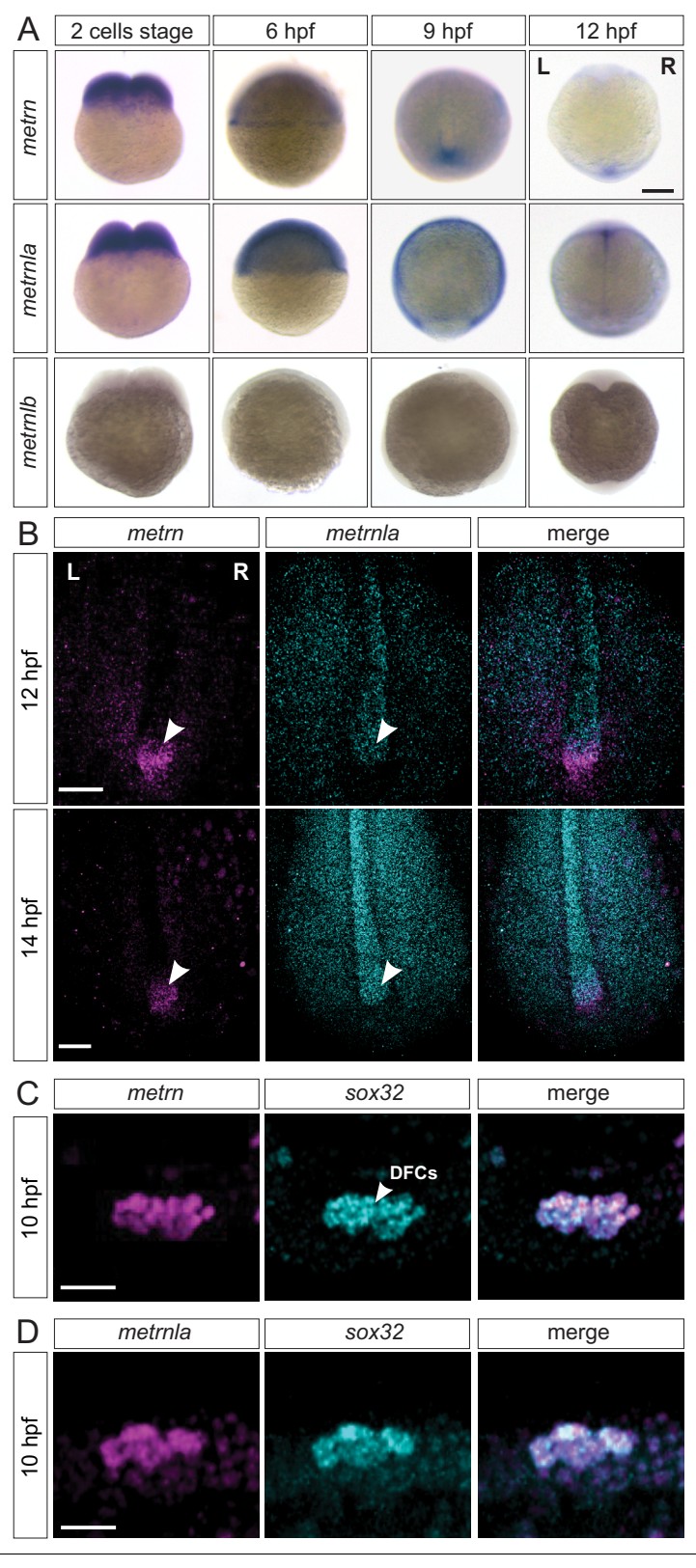

**Figure 3.** *Metrns* are expressed during early zebrafish development. (**A**) Expression patterns of *metrn*, *metrnla*, and *metrnlb* during early embryonic development from two-cell stage to 12 hpf. (9 hpf and 12 hpf dorsal view) (**B**) Confocal cross-section of the midline region of 12 and 14 hpf zebrafish embryos showing mRNA expression of the *metrn* (magenta) and *metrnla* (cyan) and their co-expression in the area of the Kupffer's vesicle (KV, indicated

*Figure 3 continued on next page*

*Figure 3 continued*

by arrowheads) by HCR. (**C**) *Metrn* and *sox32* are co-expressed by dorsal forerunner cells (DFCs) as shown by double fluorescence in situ hybridization against *metrn* (magenta) and *sox32* (cyan) on 10 hpf wild type embryos. (**D**) *Metrnla* and *sox32* are co-expressed by DFCs as shown by double fluorescence in situ hybridization against *metrnla* (magenta) and *sox32* (cyan) on 10 hpf wild type embryos. Scale bars in (**A**): 500 μm, (**B**) 250 μm, (**C**) and (**D**): 50 μm. L=left, *R*=right.

The online version of this article includes the following figure supplement(s) for figure 3:

**Figure supplement 1.** *Metrns* expression during early zebrafish and chick development.

complete colocalization of *metrn*/*sox32* and *metrnla*/*sox32* signals at 10 hpf, indicating that *metrns* genes are expressed in DFCs, the cells that will later form the KV (*Figure 3C–D*). To assess whether *metrn* and *metrnla* expression in the KV structure is conserved among vertebrates, we performed in situ hybridization for *metrn* and *metrnla* in the chick embryo at early stages. Around stage HH6, we could observe *metrn* and *metrnla* expression at the Hensen's node and around the primitive streak (marked by primitive streak landmark *fgf8*; *Figure 3—figure supplement 1D*). Remarkably, the chick *metrn* and *metrnla* expression patterns were reminiscent of the ones that we observed in zebrafish, suggesting that *metrn* gene expression at early developmental stages in embryonic Node and midline structures is conserved throughout vertebrates.

## *Metrns* loss-of-function leads to DFC disorganization and migration defects

The specific expression of *metrn* and *metrnla* in DFCs and in the KV area at early developmental stages led us to ask what role they may play for these cells and the KV formation and function. To assess whether the loss of Metrn protein function could affect the migratory and clustering properties of DFCs, we labeled these cells in 9 hpf (80–90% epiboly) gastrulating mutant embryos using the DFC landmarks *sox32* (*Alexander et al., 1999*; *Kikuchi et al., 2001*), *tbxta* (*Amack and Yost, 2004*) or *sox17* (*Alexander et al., 1999*). Compared to wild type embryos displaying ovoid DFC clustering, about 65% of the triplMut embryos displayed multiple segregated DFC clusters or abnormal linear cellular distributions as shown using *sox32* marker in 9 and 10 hpf triplMut embryos (*Figure 4A-B*, *Figure 4—figure supplement 1B*, *Table 2*). Similarly, *metrn*⁻/⁻ as well as *metrnla*⁻/⁻ single mutant embryos presented clustering defects of *sox32*-expressing DFC compared to wild type embryos (*Figure 4B*, *Figure 4—figure supplement 1A-B*, *Table 2*). In single or triplMut⁺/⁻ heterozygous embryos, the DFC clustering was instead not significantly or very mildly affected as revealed (*Figure 4—figure supplement 1B*, *Table 2*). The observed defects were also present when using *tbxta* and *sox17* as DFC markers whose expression was detected at 8 and 10 hpf respectively, indicating that although disorganized, the DFC identity was not altered by the absence of *metrns* expression (*Figure 4—figure supplement 1C*). Together, these results indicate that Metrns are required for the proper clustering of DFCs.

It was reported that proper adherent cell assembly is crucial for DFCs directional collective cell migration during their animal to vegetal pole (AP to VP) journey (*Matsui et al., 2015*; *Figure 4C*). We therefore evaluated the migration capacity of DFCs during early development in the absence of Metrn proteins. To do so, we first labeled *sox32*-expressing DFCs by in situ hybridization in wild type and triplMut embryos and measured the position of *sox32*-expressing DFCs along the AP-VP axis divided by the total embryo length at 6 hpf, 8 hpf, and 10 hpf. Compared to wild type embryos, DFCs in triplMut embryos showed migratory delays of about 8% to 13% at all analyzed developmental stages (*Figure 5—figure supplement 1B–C*, *Table 3*). By live imaging, we tracked the in vivo migration of *sox17*:GFP-labeled DFCs both in triplMut and wild type embryos from 50% epiboly to 90% epiboly (from 6 hpf; *Figure 4D*, *Figure 5—figure supplement 1D*, *Figure 4—videos 1; 2*). In line with the DFC clustering abnormalities in fixed triplMut embryos that we observed, the tracking of these cells in triplMut embryos *versus* wild type control embryos revealed a diminished convergence rate for triplMut DFCs (*Figure 4E*, upper panel). Furthermore, we measured a significant decrease in their migration speed compared to the wild type DFCs (*Figure 4E*, lower panel). Intriguingly, individually tracked DFCs exhibited a consistent movement toward the vegetal pole, regardless of the absence of Metrns proteins (*Figure 4D*, *Figure 5—figure supplement 1D*, *Figure 4—videos 1; 2*). Taken together, these findings indicate that the absence of Metrn proteins does not affect DFC migration

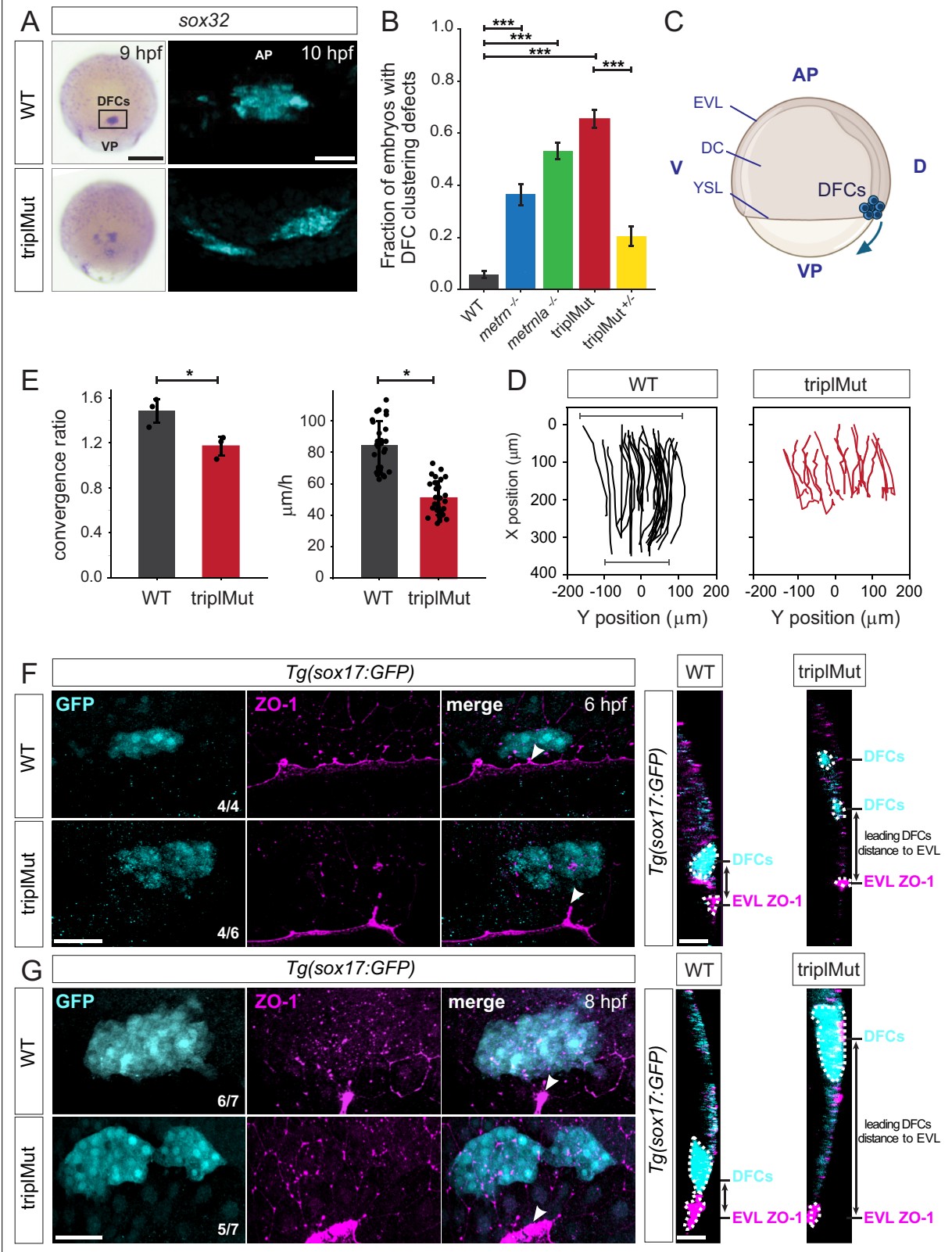

**Figure 4.** *Metrns* loss-of-function leads to DFC clustering and migration defects. (**A**) Dorsal views of *sox32* expression in DFCs at 9 hpf and 10 hpf (cyan) in wild type (WT) and triplMut embryos reveal DFC misclustering in triplMut embryos. (**B**) Quantification in fraction of embryos with DFC clustering defects at 9 hpf in WT, *metrn*−/−, *metrnla*−/−, triplMut, and triplMut+/− embryos (Fisher exact test, ***p-value: 1.6e-07 for WT vs. *metrn*−/−; ***p-value: 3.5e-40 for WT vs. *metrnla*−/−; ***p-value: 1.5e-49 for WT vs. triplMut and ***p-value: 7.9e-15 for triplMut vs. triplMut+/−). Error bars indicate standard deviation.

*Figure 4 continued on next page*

*Figure 4 continued*

(**C**) Lateral view of a schematic representation of an 8 hpf zebrafish embryo visualizing animal pole (AP) to vegetal pole (VP) dorsal forerunner cell (DFC) migration from both poles. Created with BioRender.com. (**D**) Tracking plots of combined DFC movement in wild type (left) and triplMut embryos (right) (n=3 embryos per condition, single embryo traces see *Figure 5—figure supplement 1D*) showing directed DFC movement in both conditions. Bars indicating the analyzed stretch of tracked cells along the y-axis at the beginning and end of each recording. (**E**) Plots for convergence (upper) and migration speed (lower) analyzed from DFC tracking data show a significant decrease in these parameters in triplMut embryos compared to WT controls. (*p-value convergence ratio: 0.0295; ***p-value for migration speed 1.217e-04) Error bars indicate standard deviation.(**F–G**) GFP and ZO-1 immunostainings of the dorsal margin and confocal microscopy ZY-planes (right panels) of *Tg(sox17:GFP)* WT and triplMut embryos at (**F**) 6 hpf and (**G**) 8 hpf (shield stage and 75% epiboly) showing the apical domains of marginal DFCs with ZO-1 enriched junction points (arrowheads) and revealing the absence of apical ZO1-enrichment and detachment from the EVL/YSL in triplMut embryos. EVL, enveloping layer; DC, deep cells; YSL, yolk syncytial layer; D, dorsal; V, ventral. Scale bars in (**A**): 50 μm, (**F–G**): 50 μm. AP = animal pole, VP = vegetal pole.

The online version of this article includes the following video and figure supplement(s) for figure 4:

**Figure supplement 1.** *Metrns* loss-of-function leads to DFC disorganization and migration defects.

**Figure 4—video 1.** In vivo imaging of GFP-positive DFCs (single cells labeled with *H2B-mRFP* mRNA, full stack) in a WT (WT 2 from *Figure 5—figure supplement 1D*) embryo from 50% epiboly for 3 hr showing directed DFC movement and clustering.
https://elifesciences.org/articles/105430/figures#fig4video1

**Figure 4—video 2.** In vivo imaging of GFP-positive DFCs (single cells labeled with H2B mRFP mRNA, full stack) in a triplMut embryo (triplMut 2 from *Figure 5—figure supplement 1*) from 50% epiboly for 3 hr showing directed DFC movement, but clustering defects.
https://elifesciences.org/articles/105430/figures#fig4video2

directionality but alters instead their migratory abilities, including migration speed and convergence rate.

It was shown that during the process of epiboly, initial EVL cells destined to develop into DFCs exhibit an accumulation of ZO-1 along their apical junctions. Subsequently, their apical surface gradually diminishes in size, resulting in the formation of discrete apical domains enriched with ZO-1 (*Oteíza et al., 2008*; *Pulgar et al., 2021*). DFCs as such originate from dorsal EVL cells through a mechanism involving the delamination of epithelial cells, facilitated by apical constriction. We asked whether the observed DFC organization and migration disruptions in the absence of Meteorin proteins might be attributed to perturbations in both ZO-1 accumulation and apical constriction disruption. To explore this hypothesis, we examined the spatial arrangement of these regions enriched with ZO-1 in triplMut embryos and WT controls during epiboly at 6 hpf and 8 hpf. As expected, immunohistochemistry against ZO-1 on 6 and 8 hpf *Tg(sox17:GFP)* wild type control embryos where DFCs are labeled with GFP was enriched in ZO-1 at their apical side and apical junctions (*Figure 4F-G*, *Figure 5—figure supplement 1E*). At this same developmental stage, a reduced ZO-1 enrichment at the apical junctions of triplMut GFP-positive DFCs could be detected (*Figure 4F*, *Figure 5—figure supplement 1E* left plot). Similarly, at 8 hpf, GFP-positive triplMut DFCs exhibited diminished ZO-1 at their apical region (*Figure 4G*, *Figure 5—figure supplement 1E* right plot). Furthermore, compared to wild type DFCs, triplMut GFP-positive DFCs appear to have lost their apical attachments to the EVL (*Figure 4F–G*, right panels). Indeed, the distance measured between triplMut DFCs and the EVL/YSL at 6 and 8 hpf was significantly higher compared to the one of wild type DFCs (6 hpf: *p-value: 0.039, N=4 WT, 30 DFCs measured, triplMut N=6, 46 DFCs measured; 8 hpf: *p-value: 0.021, N=7 WT embryos, 47 DFCs measured, N=7 triplMut, 54 DFCs measured; *Figure 5—figure supplement 1F*).

Together, our results demonstrate that triplMut DFCs migrate significantly slower and converge less compared to wild type DFCs. Furthermore, in the absence of Metrns, DFCs lose apical ZO1-enrichment

**Table 2.** Quantification in percentage of embryos with DFC clustering defects at 9 hpf in WT, *metrn*⁻/⁻, *metrnla*⁻/⁻, triplMut, and triplMut⁺/⁻ embryos.

| | WT-like (% of embryos) | DFC clustering defects (% of embryos) |
|---|---|---|
| WT (N=202) | 93.07 | 6.93 |
| *metrn*⁻/⁻ (N=140) | 63.57 | 36.43 |
| *metrnla*⁻/⁻ (N=245) | 46.94 | 53.06 |
| triplMut (N=182) | 34.62 | 65.38 |
| triplMut⁺/⁻ (N=140) | 79.51 | 20.49 |

**Table 3.** Quantification of the DFC migration calculated as the percentage of the total embryo length at 6, 8, and 10 hpf in wild type (WT) *vs.* triplMut embryos.

|  | 6 hpf | 8 hpf | 10 hpf |
|---|---|---|---|
| WT | 59.32% (+/-7.65%, N=27) | 78.18% (+/-7.94%, N=19) | 93.67% (+/-2.68%, N=16) |
| triplMut | 46.26% (+/-9.83%, N=24) | 69.94% (+/-6.22%, N=24) | 80.67% (+/-6.56%, N=20) |

and attachment to the EVL/YSL, indicating that Metrn proteins are important for proper DFC clustering, migration along the AP-VP axis, and their loss affects ZO-1 accumulation for the connection to the EVL/YSL.

## *Metrns* loss-of-function impairs the KV formation and function

At the end of gastrulation, DFCs that migrated to deeper layers of the developing zebrafish embryo form a lumen to shape the KV, a ciliated organ transiently present (from around 10.5 hpf to 30 hpf *Kimmel et al., 1995*) that is crucial for initiating L-R asymmetric patterning (*Essner et al., 2005*; *Cooper and D'Amico, 1996*). As we observed at 14 hpf, it was reported that Dand5, a member of the Cerberus/Dan family of secreted factors, is expressed adjacent to the KV with a right – left (*R>L*) bias between 11 and 14 hpf in zebrafish (*Figure 2*; *Hashimoto et al., 2004*). The right-biased expression of *dand5* was shown to be directly influenced by the direction and the strength of the Nodal flow generated by the KV (*Sampaio et al., 2014*). In return, Dand5 contributes to the left-biased expression of *spaw* in the LPM by antagonizing Spaw activity (*Hashimoto et al., 2004*). The leftward-directed flow generated by the KV was also shown to be important for the asymmetric expression of other Nodal genes, like *spaw*, *lefty1* (*lft1*), and *lefty2* (*lft2*) (*Hirokawa et al., 2006*; *Harvey, 1998*). The observed disturbed expression of these Nodal factors in the absence of Metrns led us to hypothesize that Metrn proteins might not only be critical for the proper clustering and migration of DFCs but also for the proper formation and function of the KV. To assess this question, we first labeled the cilia present in the KV of wild type and triplMut embryos at 14 hpf by anti-acetylated tubulin immunostaining (*Figure 5A*). We then measured the number of cells in the KV, the KV diameter, and the number of cilia, which were correlating with the number of cells, of both wild type and triplMut embryos (*Figure 5B-D*, *Figure 5—figure supplement 1*). Previous work has shown that the KV diameter and the number of KV cilia cells, as well as KV cilia, can be highly variable from embryo to embryo (*Gokey et al., 2016*). Nevertheless, despite the wide variability, we observed highly significant reduction in all cases (*Figure 5B–D*). In addition, compared to the wild type controls, triplMut embryos also displayed a significant reduction in the KV cilia length (*Figure 5E*).

Furthermore, we investigated both the KV lumen and the process of DFC-to-KV organization. As described recently and observed in ZO-1 stained embryos, by 14 hpf, DFCs clustering into a rosette structure lead tight junctions to form the KV lumen (*Oteíza et al., 2008*; *Figure 6—figure supplement 1A*, left panel). In contrast, ZO-1 immunostained triplMut embryos of the same developmental stage displayed an irregular ZO-1 lattice pattern and disrupted KV lumen shape (*Figure 6—figure supplement 1A*, right panel). Additionally, unlike in control embryos, triplMut embryos displayed only partially polarized DFCs indicated by absent or irregular aPKC-ζ immunolabeling (*Figure 6—figure supplement 1B*). These results indicate that the loss of Metrns function also impacts proper DFC polarization and subsequent KV formation.

We next evaluated whether the deficient formation of the lumen, as well as the reduced number and cilia length in triplMut, altered the KV function. To do so, we compared the Nodal flow generated by the KV of triplMut embryos to the one of wild type embryos. The movement generated by the KV cilia was monitored by injecting fluorescent microspheres (0.5 μm) into the KV at 12–14 hpf. To quantify the flow inside the KV, fluorescent microbeads movement was tracked in WT control (n=363 beads in 4 embryos) and in triplMut embryos (n=318 beads in 4 embryos; *Figure 5F*). Beads movement analysis within the KV showed a counter-clockwise rotation of the flow in wild type control (*Figure 5F*, *Figure 5—video 1*). In contrast, this movement was severely impaired in triplMut embryos (*Figure 5F*, *Figure 5—video 2*). To quantify the properties of the beads' displacement, we computed the mean square displacement (MSD), which enables us to distinguish between directed and confined motion. This analysis revealed a directed motion of microbeads in the wild type KV (MSD quadratic

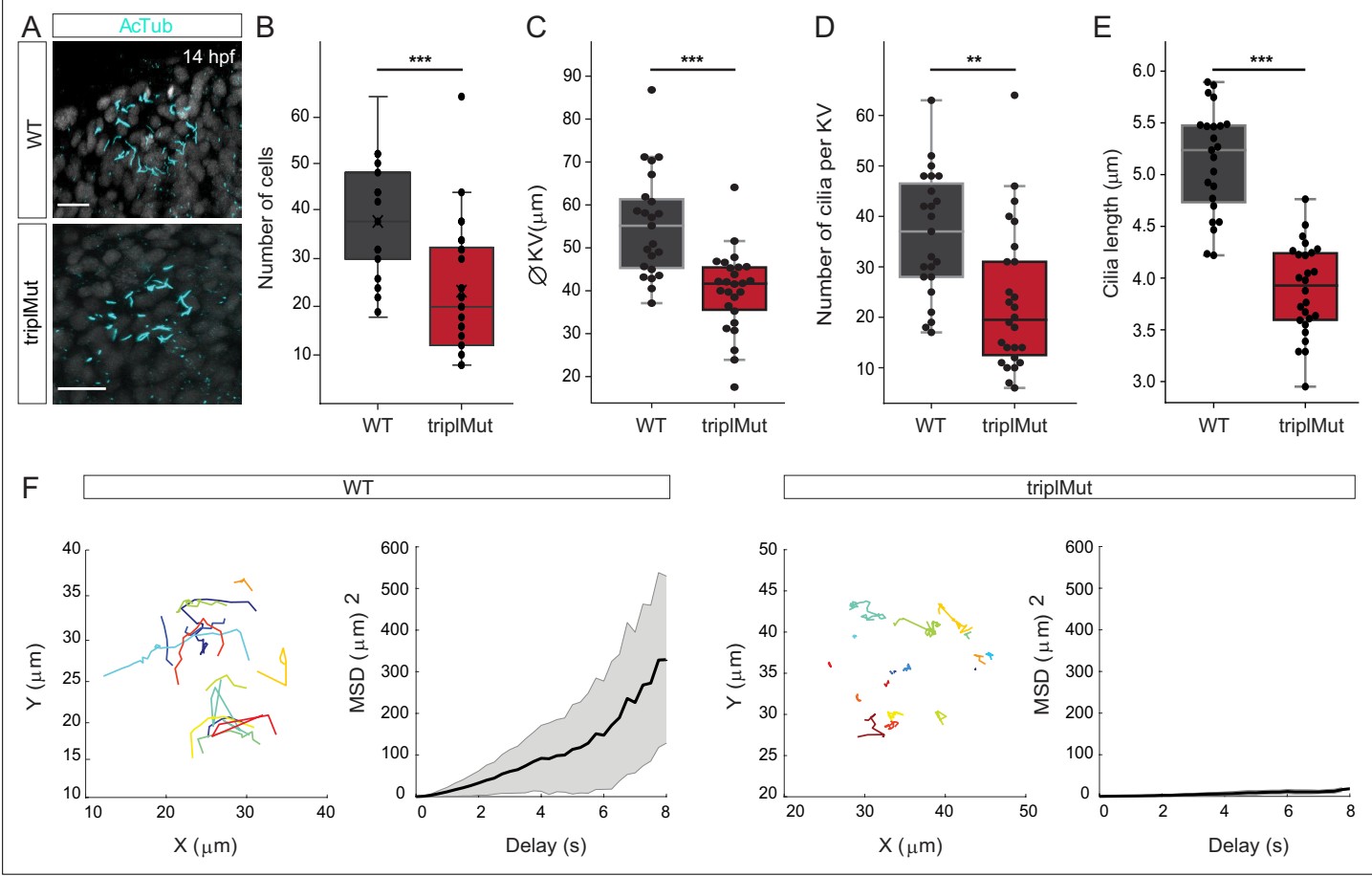

**Figure 5.** *Metrns* loss-of-function impairs Kupffer's vesicle formation and function. (**A**) Confocal cross-section of 14 hpf wild type (WT) and triplMut counterstained with acetylated tubulin (AcTub) labeling the cilia of the Kupffer's vesicle (KV). (**B**) The quantification of the cell number per KV in WT and triplMut embryos at 14 hpf shows a significant difference. (**C**) The KV diameter size measurement, (**D**) cilia number per KV quantification, and (**E**) individual cilia length measurement in WT and triplMut embryos at 14 hpf are all significantly decreased in triplMut (N WT: 23, N triplMut: 26, Student t-tests, ***p-value: 0.00028 for KV cell number, ***p-value: 9.7e-06 for KV diameter size, **p-value: 0.0012 for cilia number. For cilia length: average per KV and ***p-value: 5.8e-12). All error bars indicate standard deviation.(**F**) Single microbeads tracking in the KV of 12 hpf –14 hpf WT and triplMut embryos and the mean square displacement (MSD), revealing a directed trajectory in WT samples in contrast to triplMut, displaying short and undirected trajectories. Scale bar in (**A**): 20 µm.

The online version of this article includes the following video and figure supplement(s) for figure 5:

**Figure supplement 1.** *Metrns* loss-of-function leads to DFC disorganization and migration defects.

**Figure 5—video 1.** Microbeads tracking the Kupffer's vesicle of 12–14 hpf wild type embryos showing a distinctive anti-clockwise rotation.
https://elifesciences.org/articles/105430/figures#fig5video1

**Figure 5—video 2.** Microbeads tracking the Kupffer's vesicle of 12–14 hpf triplMut embryos showing the loss of the distinctive anti-clockwise rotation.
https://elifesciences.org/articles/105430/figures#fig5video2

dependence on Δt) and a confined motion (MSD asymptotic behavior) in triplMut (*de Bruin et al., 2007*). Accordingly, measurement of average bead velocity showed an overall reduction in the beads' speed in the triplMut mutant embryos compared to the wild types (WT mean = 0.96212 µm/s; TriplMut mean = 0.47996 µm/s; *Figure 6—figure supplement 1C*). These results demonstrate that the observed defects in the KV formation in the absence of Metrns also severely impact its function and thereby the Nodal flow generation, responsible for proper Nodal factors expression.

## Genetic interaction of ItgαV/Itgβ1b and Metrns on the level of the DFCs

In order to further uncover the mechanisms through which Metrn proteins act in the establishment of early L-R patterning, we finally assessed the possible relationship between Metrns and mediators of the extracellular matrix like Integrin αV1 (ItgαV1) and Integrin β1b (Itgβ1b). Integrins are membrane receptors involved in cell adhesion and recognition during embryogenesis, amongst others (*Hynes, 1992*). During gastrulation, *itgαV* as well as *itgβ1b* were shown to be expressed in DFCs, and *itgαV* and *itgβ1b* morpholino-injected embryos display DFC phenotypic defects reminiscent of those observed in *metrn* mutants (*Ablooglu et al., 2010*).

We therefore asked whether Metrns loss of function could affect the expression of both integrins. To do so, we quantified *itgαV* and *itgβ1b* expression levels in wild type and triplMut embryos at 6, 9, and 24 hpf by qRT-PCR. No significant change in *itgαV* and *itgβ1b* expression levels at either tested developmental stage was measured in triplMut embryos compared to wild type controls (*Figure 6— figure supplement 2A*).

In order to test whether ItgaV1 and Itgβ1b and Metrns genetically interact, we injected insufficient doses of morpholinos for *itgβ1b* (0.5 ng) or *itgαV1* (0.41 ng) into 1–4 cell stage in a heterozygous *metrns* mutant background. The injection of the morpholinos at these doses and at sufficient doses (*itgβ1b:* 0.7 ng, *itgαV1:* 1.25 ng) reproduced the previously reported defects on DFCs (*Figure 6A*, *Table 4*; *Ablooglu et al., 2010*). Instead, the injection of low doses of either *itgβ1b* or *itgαV1* morpholino did not affect DFC clustering, similarly to non-injected triplMut$^{+/-}$ embryos (*Figure 6A*, *Table 4*). However, a significantly higher percentage of embryos with DFC clustering defects could be observed upon the injection of either morpholino at low dose into triplMut$^{+/-}$ (*Figure 6A*, *Table 4*). These results are consistent with the hypothesis that Metrn proteins act together with both Integrins for the proper clustering and migration of DFCs during early embryonic development without directly affecting their expression.

## Discussion

Proteins of the Meteorin family were first described as secreted neurotrophic factors highly conserved among vertebrates (*Kim et al., 2014*; *Nishino et al., 2004*; *Ramialison et al., 2008*; *Jørgensen et al., 2012*). In mice, *Metrnla* mutants display phenotypes as impaired angiogenesis after ischemic injury (*Reboll et al., 2022*) or dysregulated cytokine production (*Ushach et al., 2018*). The disruption of *Metrn* resulted in early embryonic lethality (*Kim et al., 2014*), preventing the investigation of the combined role of Metrn proteins in early embryonic development in vivo and, in particular, its role in L-R patterning. Here, using the zebrafish larva as a model system, we generated knockout lines for all three zebrafish *metrn* genes, *metrn*, *metrnla,* and *metrnlb*, which are all viable and fertile, possibly attributed to species-specific differences in compensatory mechanisms, allowing for the study of Meteorins functions during early embryonic development.

Our results demonstrate that Metrns are required for symmetry breaking and the establishment of L-R asymmetry, a novel role for Meteorin proteins during vertebrate embryonic development. All three paralogs analyzed, when genetically deleted, had an effect on heart looping formation, with the strongest effect for Metrnla and Metrn, while Metrnlb had a smaller but still significant effect. In our in situ analysis, we failed to detect early *metrnlb* expression. We do not exclude that at early stages, *metrnlb* could be expressed at very low levels or in a very small number of cells that we cannot detect. However, in the *metrnlb* mutant, the loss of this expression could still be sufficient to induce the low-penetrant phenotype. Interestingly, when analyzing the expression of these genes in a single cell transcriptomic atlas at the earliest available time point (10 hpf), in contrast to the robust expression of the two first genes, *metrnlb* expression is indeed found in very few cells (*Kim et al., 2024*).

We show that Metrns play a crucial role in ensuring this function, the genetic interaction with the integrins ItgαV1 and Itgβ1b, known to facilitate the proper clustering and migration of DFCs based on the sufficient accumulation of ZO-1 and apical constriction, essential to form the KV (*Oteíza et al., 2008*; *Ablooglu et al., 2010*). Furthermore, our results reveal that Metrns are essential for several aspects of the KV formation and function, including the cilia length, cilia number, KV size, and lumen formation. Consequently, Metrns are important for the regulation of the proper expression of Nodal

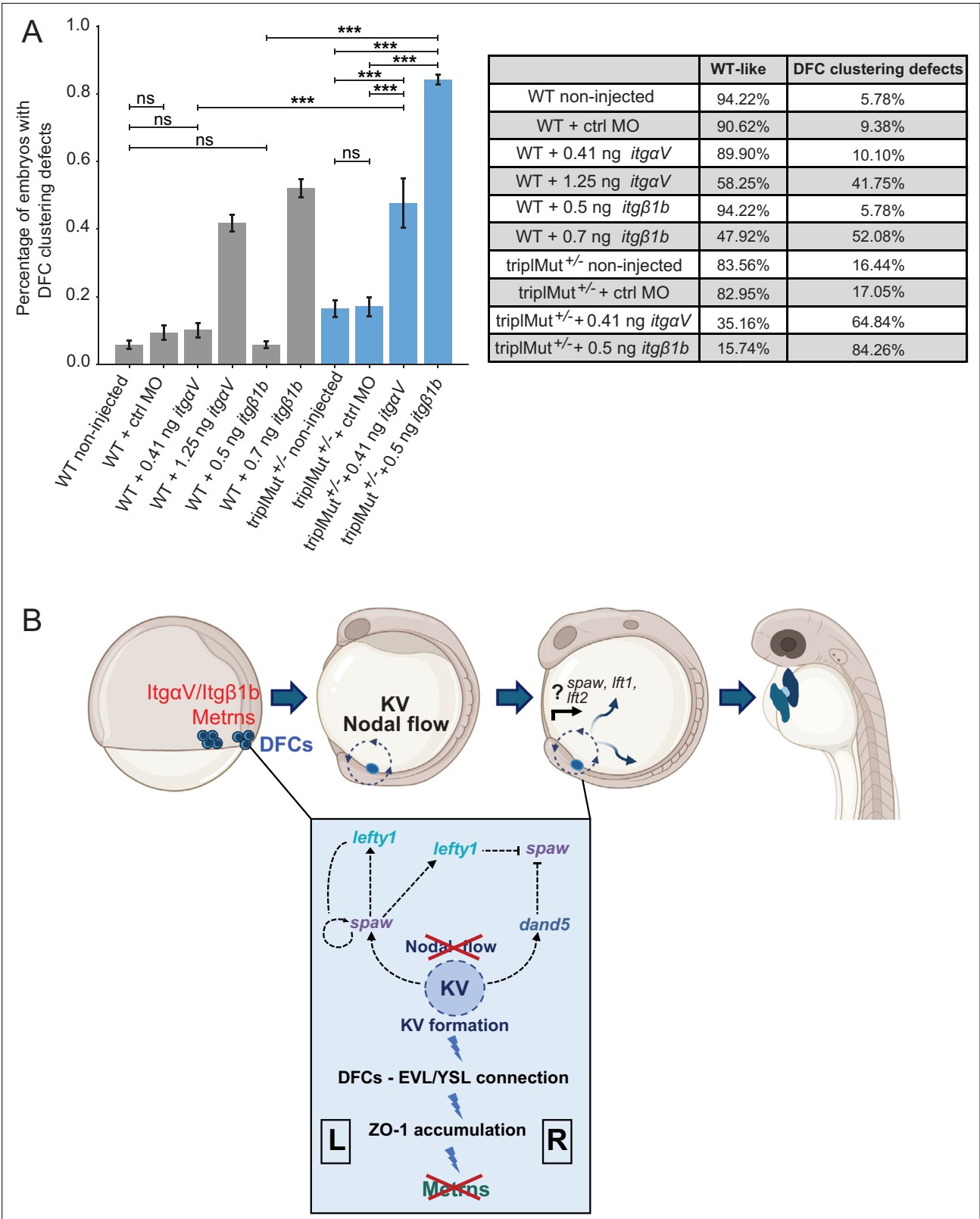

**Figure 6.** Metrns and ItgαV/Itgβ1b act together for the proper clustering and migration of DFCs. (**A**) Quantification in percentage of embryos with DFC clustering defects at 9 hpf in non-injected wild type (WT), WT +2.5 ng control morpholino (WT +ctrl MO), WT +insufficient doses of *itgαV/itgβ1b* MOs (WT +0.41 ng *itgαV*; WT +0.5 ng *itgβ1b*), WT +sufficient *itgαV/itgβ1b* MO doses (WT +1.25 ng *itgαV*; WT +0.7 ng *itgβ1b*), triplMut⁺/⁻ non-injected, triplMut⁺/⁻ + 2.5 ng ctrl MO and triplMut⁺/⁻ with insufficient doses of *itgαV/itgβ1b* MOs (triplMut⁺/⁻ + 0.41 ng *itgαV*; triplMut⁺/⁻ + 0.5 ng *itgβ1b*).

*Figure 6 continued on next page*

*Figure 6 continued*

Fisher exact tests indicate *p-value <0.01, **p-value <0.001 and ***p-value <0.0001 and ns = not significant. Error bars indicate standard deviation. (**B**) Schematic representation modeling the novel role of Metrns for DFCs assembly and migration, for the proper KV formation and subsequent Nodal factors expression distribution at the base of the correct L-R axis establishment during early development. Created with BioRender.com.

The online version of this article includes the following source data and figure supplement(s) for figure 6:

**Figure supplement 1.** *Metrns* loss-of-function impairs Kupffer's vesicle formation and function.

**Figure supplement 2.** *Kita, kitb,* and *htr2b* expression patterns during early zebrafish development.

**Figure supplement 2—source data 1.** qRT-PCR values for itgav and itgb1b expression levels at 6 hpf, 9 hpf and 1 dpf in wt and triplMut backgrounds.

flow genes, and this regulation is critical for the proper asymmetric positioning and morphology of vital organs like the heart (*Figure 6B*).

Previous work has shown that zebrafish midline mutants display altered *lft1* and *lft2* expression patterns as well as discordance among heart, gut, and brain gene expression patterns (*Bisgrove et al., 2000*). From 9 hpf, *metrn* expression was visible around the developing LPM close to the midline and the leading edge of the shield. *Metrnla* expression was found throughout the whole enveloping layer and the developing midline, which plays an important role in vertebrate L-R development. Despite the expression of both paralogs close to or at the midline, we hypothesize that the described body asymmetry defects in triplMut embryos do not originate from impaired midline structures but rather impaired DFC clustering and migration. Indeed, triplMut<sup>-/-</sup> embryos display intact *tbxta*-positive notochords (*Figure 4—figure supplement 1C*), as well as randomized visceral organs positioning (*Figure 1D–E*), phenotypes that are not characteristic for defects of the anterior midline (*Bisgrove et al., 2000*).

DFCs are precursors of the zebrafish KV, organ of laterality, essential for the L-R axis establishment. DFCs develop at the beginning of gastrulation and form the KV by the end of gastrulation (*Essner et al., 2005*; *Cooper and D'Amico, 1996*). Given the early onset of *metrn* and *metrnla* expression and their specific expression within the DFCs, we investigated their role in the clustering and migration capacity of DFCs during these early embryonic developmental steps. In the course of their migration, DFC progenitors intercalate mediolaterally and result in an oval-rosette shape cluster of DFCs by mid-gastrulation that will form the KV (*Oteíza et al., 2008*). *Metrn<sup>-/-</sup>, metrnla<sup>-/-</sup>,* and triplMut zebrafish embryos displayed DFC assembly defects and delayed migration of these cells during gastrulation. Since DFCs in triplMut still express DFC-specific markers (*sox17, sox32, tbxta*), we concluded that Metrns are not required for the specification of DFCs per se, but rather that Metrns are important for their proper clustering and migration.

It is important to note that the observed DFC clustering and migration abnormalities in Metrn mutants primarily affect the apico-ventral axis rather than the mediolateral axis. This is evident from the disrupted connection between DFCs and the YSL/EVL in these mutants, while the migration

**Table 4.** Number of embryos with DFC clustering defects at 9 hpf upon *itgβ1b*, *itgαV* or control (ctrl) morpholino (MO) injection in wild type (WT) or triplMut<sup>+/-</sup> embryos.

| Experiment | Number of embryos |
|---|---|
| WT non injected | 346 |
| WT +ctrl MO | 192 |
| WT +0.41 ng *itgαV* MO | 208 |
| WT +0.5 ng *itgβ1b* MO | 623 |
| WT +1.25 ng *itgαV* MO | 412 |
| WT +0.7 ng *itgβ1b* MO | 336 |
| triplMut<sup>+/-</sup> non injected | 219 |
| triplMut<sup>+/-</sup> + ctrl MO | 176 |
| triplMut<sup>+/-</sup> + 0.41 ng *itgαV* MO | 182 |
| triplMut<sup>+/-</sup> + 0.5 ng *itgβ1b* MO | 559 |

directionality remains intact. Since a proper connection between DFCs and the EVL facilitates a subset of DFCs to establish apical contacts with the EVL and become pre-polarized, one could speculate that Metrns are involved in the polarization and establishment of direct apical contacts of DFCs with the EVL (*Oteíza et al., 2008*; *Cooper and D'Amico, 1996*; *Pulgar et al., 2021*; *Solnica-Krezel et al., 1996*). This disrupted connection in *metrns* loss-of-function embryos, which usually links DFCs with the vegetal spread of extra-embryonic tissues (*Pulgar et al., 2021*), likely results in the observed DFCs vegetal motion defects.

It has been demonstrated that these apical attachments function as tissue connectors, linking DFCs with the spreading of extra-embryonic tissues towards the vegetal pole (*Pulgar et al., 2021*).

Therefore, the loss of apical attachments in the mutant embryos causes all DFCs to converge less and migrate more slowly than in wild-type controls, as shown by our in vivo imaging analysis.

This would imply that a number of DFCs was simply too slow to reach the vegetal pole in time to form a fully functional KV. Given that KV cilia are monocilia, this subsequently results in a diminished count of KV cells (*Figure 5—figure supplement 1A*), along with the observed structural abnormalities in the KV lumen and impairments in its overall functionality.

Additionally, it was shown that the formation of the developing KV lumen was directly connected to ciliogenesis as polarized DFCs initiate developing cilia at their membrane domain facing the forming lumen (*Oteíza et al., 2008*). This indicates that proper DFC polarization is essential for KV formation and for ciliogenesis. In *metrns* mutants, the reduced cilia length and the lack of PKC $\zeta$ staining in the developing KV further highlight an impairment of DFC polarization in the absence of Metrns, consequently leading to KV formation and function defects. Further work would be needed to explore whether Metrns directly impact the motility of KV cilia or if the disrupted Nodal flow observed is solely caused by the disturbed KV lumen.

It has been demonstrated that appropriate asymmetrical Nodal flow plays a crucial role in the asymmetrical expression of Nodal genes such as *dand5, spaw, lefty1 (lft1)*, and *lefty2 (lft2)* (*Hirokawa et al., 2006*; *Harvey, 1998*; *Hashimoto et al., 2004*). As such, the disrupted formation and function of the KV in *metrns* mutants might contribute to the evident perturbation in Nodal factor expression and may also induce randomization of Nodal factor expression in L-R asymmetry (*Essner et al., 2005*; *Nonaka et al., 1998*; *Gokey et al., 2016*; *Ablooglu et al., 2010*). We do not exclude, however, that Metrns directly impact the expression of *spaw, leftys,* and *dand5* in addition to their role in DFC cell behaviors and KV formation. One of the earliest functions attributed to Meteorin was as a regulator of mesendoderm development through the enhancement of nodal expression (*Kim et al., 2014*). In this context, several studies have reported that Nodal signaling was shown to regulate DFC specification (*Oteiza et al., 2010*; *Choi et al., 2007*).

In line with the impaired KV formation and function that we observed in *metrns* mutants, it was shown that knockdown of Integrin αV as well as Integrin β1b in zebrafish results in L-R asymmetry defects (*Ablooglu et al., 2010*). This was explained by DFC organization and migration defects and disturbed KV formation. IntegrinαVβ1 specific functions have not yet been completely resolved in mammals, mostly due to their broad expression and redundant function with other integrins, as well as the lack of antibodies and inhibitors specific for αVβ1. However, several in vitro studies have already connected αVβ1 to developmental processes. For instance, αVβ1 was shown to be implicated in embryonic astrocytes and oligodendrocyte precursors migration in rodents (*Milner et al., 1996*) and neural cell adhesion molecule L1 cell binding (*Felding-Habermann et al., 1997*). Additionally, the fibronectin receptor integrin α5β1 has been found to be crucial for regulating extracellular matrix assembly along tissue boundaries, in coordination with Eph/Ephrin signaling (*Jülich et al., 2009*).

Our results suggest that one mechanism by which Metrns act is through their genetic interaction with Integrin αV/β1b to mediate the proper clustering of DFCs. These interactions could influence Integrin's adhesive functions (or other properties) necessary for DFC clustering. If so, the absence of Meteorins would lead to disturbed adhesion and/or Integrin signaling, resulting in DFC migration defects and disturbed KV development. Interestingly, it was shown that in αV and β1b morphants, DFC-EVL connections remain intact (*Ablooglu et al., 2010*), unlike in our *meteorin* mutants. This contrast suggests that Meteorins alone are crucial for DFC polarization and that their interaction with integrins αVβ1b may rather be important for the subsequent steps leading to KV formation. While our genetic interaction studies provide strong evidence of a link between Meteorins and Integrins, further biochemical analyses are required to fully elucidate this interaction.

Despite the recent advancements in uncovering the multiple functions of Meteorins, their receptors during early development are still unknown. It was recently shown that METRNL is promoting heart repair through its binding to the KIT tyrosine kinase receptor, establishing METRNL as a KIT receptor ligand in the context of ischemic tissue repair (*Reboll et al., 2022*). In another study, it was reported that Metrn is a ligand for the 5-hydroxytryptamine receptor 2b (Htr2b; *Dai et al., 2022*). Additionally, Metrn was associated with the regulation of reactive oxygen species levels in hematopoietic stem/progenitor cells via directing phospholipase C signaling (*Dai et al., 2022*). When we analyzed the expression of *kita*, one of the two zebrafish orthologs to KIT tyrosine kinase receptor, at 9 hpf, we found that *kita* is solely expressed in the prechordal plate and later (from 11 hpf) in the lateral borders of the anterior neural plate (*Figure 6—figure supplement 2B*, left panel). Its paralog *kitb* is expressed from 11 hpf in the anterior ventral mesoderm (*Figure 6—figure supplement 2B*, right panel), as previously reported (*Mellgren and Johnson, 2005*). Additionally, the analysis of *htr2b* expression during zebrafish development revealed that its expression only starts from around 2 dpf in the heart (*Figure 6—figure supplement 2C*). The absence of *kita/b* as well as *htr2b* expression at early embryonic stages in or at proximity of DFCs and in the forming KV suggests that Metrns rather act through other receptor(s) to ensure their L-R patterning functions.

Mechanisms underlying L-R patterning are known to be highly conserved within vertebrates. Hence, we raised the question of whether the Meteorin protein family is conserved across different species. Genes coding for *metrn* and *metrnl* exist in several vertebrates classes but could not be found in the genomes of invertebrates (*Zheng et al., 2016*). *Metrnl* ortholog genes could instead be found in amphioxus and tunicates. Therefore, we propose that Metrns function in the establishment of the L-R axis that we have uncovered here emerged during Chordate evolution (*Figure 6—figure supplement 2D*).

# Materials and methods

## Key resources table

| Reagent type (species) or resource | Designation | Source or reference | Identifiers | Additional information |
|---|---|---|---|---|
| Strain, strain background (*Danio rerio*, both sex) | AB wild type | Sorbonne Université | RRID:ZFIN_ZDB-GENO-960809-7 | |
| Strain, strain background (*Gallus gallus*) | Chicken fertilized eggs (JA57 strain) | EARL Morizeau | N/A | |
| Genetic reagent (*Danio rerio*, both sex) | *Tg(sox17:GFP)* | This paper | | The *pTol2-sox17:GFP* plasmid was kindly gifted from Didier Stanier and Stephanie Woo. |
| Recombinant DNA reagent | *pEGFP-sox17 (plasmid)* | Addgene | RRID:Addgene_31400 | |
| Recombinant DNA reagent | *DR274 (plasmid)* | Addgene | RRID:Addgene_42250 | |
| Peptide, recombinant protein | Phusion high-fidelity DNA polymerase | LifeTechnologies/ThermoFisher Scientific | Cat# 740609 (Thermo Fisher Scientific, RRID:SCR_008452) | |
| Peptide, recombinant protein | Taq DNA Polymerase | LifeTechnologies/ThermoFisher Scientific | Cat #10342046 (Thermo Fisher Scientific, RRID:SCR_008452) | |
| Commercial assay or kit | Megascript T7 transcription kit | Ambion | AM1334 (Ambion Inc, RRID:SCR_008406) | |
| Commercial assay or kit | mMESSAGE mMACHINE Sp6 Ultra kit | Ambion | AM1340 (Ambion Inc, RRID:SCR_008406) | |
| Commercial assay or kit | SuperScript III First-Strand Synthesis system | Invitrogen | Invitrogen 18080051 | |

*Continued on next page*

*Continued*

| Reagent type (species) or resource | Designation | Source or reference | Identifiers | Additional information |
|---|---|---|---|---|
| Commercial assay or kit | RNAeasy Mini Kit | Qiagen | Qiagen Cat no. 74104 | |
| Commercial assay or kit | Zero Blunt TOPO PCR Cloning Kit | LifeTechnologies/ThermoFisher Scientific | Cat# K280002 (Thermo Fisher Scientific, RRID:SCR_008452) | |
| Commercial assay or kit | HCR RNA-FISH (v3.0) | Molecular Instruments Inc. | HCR Buffers (v3.0) | |
| Commercial assay or kit | HCR RNA-FISH (v3.0) | Molecular Instruments Inc. | HCR Amplifier (v3.0) B1-488 B2-594 B3-647 | |
| Commercial assay or kit | RNA Labeling Kit | Roche Diagnostics Corporation | Roche Cat# 11093274910 (Roche Diagnostics, RRID:SCR_025096) | |
| Antibody | anti-acetylated tubulin | Sigma | Sigma-Aldrich Cat# T6793, RRID:AB_477585 | (1:200) |
| Antibody | anti-ZO1 | Invitrogen, Carlsbad, CA, USA | Innovative Research Cat# 33-9100, RRID:AB_87181 | (1:200) |
| Antibody | anti-GFP | GeneTex | GeneTex Cat# GTX13970, RRID:AB_371416 | (1:200) |
| Antibody | anti-aPKC | Santa Cruz Biotechnology Inc, CA, USA | Santa Cruz Biotechnology Cat# sc-216, RRID:AB_2300359 | (1:200) |
| Software, algorithm | phylogeny analysis | https://www.phylogeny.fr/ | | *Dereeper et al., 2008* |
| Software, algorithm | Matlab codes used for beads and DFC tracking | This paper | https://doi.org/10.5281/zenodo.15622175 | See Materials and methods |

## Zebrafish husbandry

Zebrafish (*Danio rerio*) were maintained at 28 °C overnight (O/N) at 14 hr light/10 hr dark cycle. Fish were housed in the animal facility of our laboratory, which was built according to the respective local animal welfare standards. All animal procedures were performed in accordance with French and European Union animal welfare guidelines with protocols approved by the committee on ethics of animal experimentation of Sorbonne Université (APAFIS#21323–2019062416186982). For all experiments in this study, primarily zebrafish AB strains were used unless stated otherwise. The *Tg*(*sox17:GFP*) lines were created via injection of a *pTol2-sox17:GFP* plasmid (kindly gifted from Didier Stanier and Stephanie Woo) into one-cell stage zebrafish embryos together with 150 ng/ml of *tol2* mRNA. Injected embryos were grown to adulthood and screened for GFP in their offspring.

## *Metrn*, *metrnla,* and *metrnlb* CRISPR-Cas9-mediated mutagenesis

SgRNA guide sequences were cloned into a Bsal-digested DR274 plasmid vector (Addgene 42250). The sgRNAs were synthesized by in vitro transcription (using the Megascript T7 transcription kit, Ambion, AM1334). After transcription, sgRNAs were purified using an RNAeasy Mini Kit (QIAGEN). The quality of purified sgRNAs was checked by electrophoresis on a 2% agarose gel. The target sequences were the following:

| | Sequence (5' - 3') |
|---|---|
| *metrn* | CCACCACACCCGGCCCAGCC |
| *metrnla* | GGTGTATCTCCGCTGCGCCC |
| *metrnlb* | CCAGAAACAGCGCCCCCTCC |

Cas9 mRNA was generated as described in *Hwang et al., 2013*. To induce targeted mutagenesis at the metrn, metrnla, and metrnlb loci, 200 ng.m L$^{-1}$ of sgRNA were injected into one-cell stage zebrafish embryos together with 150 ng.m L$^{-1}$ of cas9 mRNA. Injected embryos were grown to adulthood and screened for mutation in their offspring.

## DNA extraction and sequencing for analysis of CRISPR/Cas9-mediated mutagenesis

For genomic DNA extraction, pools of 5 dpf embryos were digested for 1 hr at 55 °C in 0.5 mL lysis buffer (10 mM Tris, pH 8.0, 10 mM NaCl, 10 mM EDTA, and 2% SDS) with proteinase K (0.17 mg/mL, Roche Diagnostics) and inactivated 10 min at 95 °C. To check for the frequency of indel mutations, target genomic loci were PCR amplified using a Phusion High-Fidelity DNA polymerase (Invitrogen). PCR amplicons were subsequently cloned into a *pCR-bluntII-TOPO* vector (Thermo Fisher). Plasmid DNA was isolated from single colonies, and Sanger sequencing was performed by Eurofins. Mutant alleles were identified by comparison with the wild type sequence using Geneious software.

## Mutants genotyping and generation of triple mutants

Genotyping of *metrns* mutants was performed as follows: after genomic DNA extraction, PCR amplification reactions were conducted in final volumes of 20 µl containing 1 x PCR reaction buffer, 1.5 mM MgCl2, 70 ng of gDNA, Taq DNA Polymerase (5 U.µ L$^{-1}$) and 0.5 µM of each primer.

|  | Sequence (5' - 3') |
| --- | --- |
| *metrn*-fw | CTGTGTTGACTGCTGGCTG |
| *metrn*-rv | GTGGTTTAGTGGTGTTCTTACAATGA |
| *metrnla*-fw | TCCCATGCCTGGACCTCATA |
| *metrnla*-rv | AGACGGAGAGAAGAGACGCT |
| *metrnlb*-fw | TGTTGATCAGCAGTGTGTGCGTAGC |
| *metrnlb*-rv | GTCCTCCGCTGATCTACGTG |

The DNA amplification was performed with 35 cycles at the annealing temperature of 60 °C. *Metrn* and *metrnlb* amplicons were loaded on a 2% agarose gel to discriminate between wild type and mutant alleles (for *metrn*, the expected size of the wild type amplicon is 598 bp while the mutant amplicon is 482 bp long using the same primer set; for *metrnlb*, the size of the expected wild type amplicon is 242 bp while the mutant band is 156 bp long using the same primer set). *Metrnla* PCR product was digested at 55 °C O/N with the PasI restriction enzyme, and the resulting digestion was subsequently analyzed on a 2% agarose gel. The wild type amplicon results in two fragments of 107 bp and 94 bp length, respectively. The mutant amplicon band size is instead 201 bp using the same primer set. Triple mutants were obtained first by incrossing homozygous *metrn* and *metrnla* mutants to generate double heterozygous mutants. Double heterozygous mutants were then crossed with homozygous *metrnlb*$^{-/-}$ fish and genotyped to identify triplMut$^{+/-}$ triple heterozygous mutants. triplMut$^{+/-}$ fish were incrossed and genotyped to identify triplMut$^{-/-}$ triple homozygous mutants. In all described experiments of this study, triplMut$^{+/-}$ were obtained by outcrossing female triplMut$^{-/-}$ fish with male wild type fish, if not stated differently to prevent from *metrns* maternal contribution.

## Hybridization chain reaction (HCR)

All HCR probes and solutions were purchased from Molecular Instruments. Dechorionated embryos at the developmental stage(s) of interest were fixed in 4% paraformaldehyde in PBS (pH 7.4) for 2 hr at room temperature (RT), followed by several washes with PBS to stop the fixation. Embryos were then dehydrated with a series of methanol (MeOH) washes and stored at –20 °C. The HCR was performed following the manufacturer's instructions. HCR samples were imaged on an inverted confocal microscope Olympus FV-1000, employing a 20 x oil immersion objective (NA 0.85). Z-volumes were acquired with a 5 µm resolution and images were processed and analyzed using ImageJ.

## In situ hybridization in zebrafish and chicken

The respective cDNA fragments were amplified by PCR from total zebrafish or chicken cDNA of stage 12 hpf – 48 hpf and HH6. In vitro transcription of Digoxigenin/Fluorescent-labeled probes was performed using an RNA Labeling Kit (Roche Diagnostics Corporation) according to manufacturer's instructions. Zebrafish and chicken whole-mount in situ hybridizations were performed as previously described (*Thisse and Thisse, 2008*; *Streit and Stern, 2001*). Stained embryos were then imaged on a Leica MZ10F stereomicroscope. Images were processed and analyzed using ImageJ software. Color balance, brightness, and contrast were applied uniformly. Fluorescently labeled samples were imaged using an inverted confocal microscope Olympus FV-1000, employing a 40 x oil immersion objective (NA 1.30). Z-volumes were acquired with a 0.5 µm resolution, and images were processed and analyzed using ImageJ.

## Analysis of DFC phenotype and migration

For the DFC clustering analysis, Bayesian inference was used to recover the maximum-likelihood value for the malformation probability. The outcome of an experiment was considered to be a binary random variable (malformed or wild type phenotype) whose probability depends solely on the experimental condition. The total number of malformed phenotypes per experiment is therefore binomially distributed. The posterior distribution is obtained by inverting this binomial distribution using Bayes theorem, under a uniform prior. Under these hypotheses, the maximum-likelihood estimate for the malformation probability is simply given by the empirical malformation frequency. The standard deviation of the posterior distribution was used to provide a confidence interval for the estimate (black error bars). p-Values were evaluated using Fisher exact test on the number of malformed and wild type phenotypes. AP to VP DFC migration was calculated as the percentage of the total embryo length.

For in vivo live DFC imaging, a *pEGFP-sox17* construct (Addgene #31400) and *H2B-RFP* mRNA were co-injected either in wild type or triplMut embryos that were embedded at 50% epiboly (6 hpf) in 1% low melting-point agarose diluted in E3 water. DFC was identified as recently described in *Pulgar et al., 2021*. Imaging was performed in a temperature constant environment with a 10 x water dipping objective using an upright Yokogawa CSU-X1 spinning disk scan head, mounted on a DM6000 upright Leica microscope and a CCD CoolSnap HQ2 camera at 4 Hz. Volumetric image stacks were acquired every 6 min which was sufficient to manually track individual DFC migration during 3 hr from 50% epiboly. No drift correction was required. The convergence ratio was calculated by comparing how far the tracked cells stretched along the y-axis at the beginning and end of each recording (as visualized in *Figure 4D* and described in *Pulgar et al., 2021*).

## qRT-PCR and analysis

For gene expression analysis, total RNA extraction was prepared from zebrafish embryos with TRIzol reagent (Thermo Fisher) and TURBO DNA-free reagents (Invitrogen AM1907M). Total RNA was cleaned up using the RNeasy Mini Kit (QIAGEN) following the manufacturer's instructions and treated twice with DNase I (1 U.µg$^{-1}$ RNA, QIAGEN). The RNA concentration was quantified using the Nanodrop2000 (Thermo Fisher). RNA (1 µg) was retro-transcribed using random primers and a SuperScript III First-Strand Synthesis system (Invitrogen 18080051). For qRT-PCR, the SYBR Green PCR Master Mix (Thermo Fisher) was used according to the manufacturer's protocol. *Ef1a* was used as a reference gene as previously reported (*Tang et al., 2007*). All assays were performed in triplicates and repeated in at least three independent experiments. The mean values of triplicate experiments were calculated according to the delta CT quantification method (*Valcu and Valcu, 2011*), and a Student T-test was applied for the p-value calculation.

## mRNA synthesis and injection

Zebrafish *metrn* and *metrnla* cDNA fragments were amplified by PCR from zebrafish cDNA of 1 dpf. The fragments were cloned into a *pCS2+* plasmid linearized with Xho1 and Xba1 restriction enzymes using a Quick Ligation Kit (NEB). The *pCS2:h2b-rfp* plasmid readily available in the laboratory was linearized with the NotI restriction enzyme, and the mRNA synthesis was performed by in vitro transcription using the mMESSAGE mMACHINE Sp6 Ultra kit (AM1340, Ambion), adding 1 µL of GTP as recommended by the manufacturer's instructions and followed by lithium chloride precipitation. 150 ng.µL$^{-1}$ of the synthesized mRNA was then injected into one- to four-cell-stage zebrafish embryos.

## Design and injection of antisense morpholino oligonucleotides

The published morpholinos (MO) used in this study for *itgαV* (5'-AGTGTTTGCCCATGTTTTGAGTCT C-3'), *itgβ1b1* (5'-GGAGCAGCCTTACGTCCATCTT AAC-3') and for control (ctrl MO) (5'-CCTCTTAC CTCAGTTACAATTTATA-3') were injected using the same reported concentration of 0.41 ng (*itgαV*) and 0.5 ng (*itgβ1b1*) for insufficient doses, 1.25 ng (*itgαV*) and 0.7 ng (*itgβ1b1*) for full knockdown effect and 2.5 ng for the ctrl MO (*Ablooglu et al., 2010*). All MOs were obtained from Gene Tools (Philomath, OR, USA). MOs were injected in 1–4 cells stage zebrafish embryos.

## Whole-mount immunohistochemistry

Embryos were fixed in 4% paraformaldehyde in PBS-Tween for 2 hr at RT and subsequently washed three times in PBS-1% Triton X-100 to promote their permeabilization. They were then incubated for 1 hr at RT in blocking solution (10% Normal Goat Serum, in PBS-Tween) followed by overnight incubation, at 4 °C with primary antibodies diluted at 1:200 in 1% blocking solution. The following primary antibodies were used: anti-acetylated tubulin (Sigma T6793); anti-ZO1 (339100, Invitrogen, Carlsbad, CA, USA); anti-GFP (GTX13970, GeneTex); anti-aPKC ζ (sc-216, Santa Cruz Biotechnology Inc, CA, USA). After five washes in 1 x PBS-Tween, the embryos were then incubated O/N with respective Alexa Fluor IgG secondary antibody (Life Technologies) diluted 1:200 in 1% blocking solution. The following day, embryos were washed five times in 1 x PBS-Tween and mounted in 1% low-melting point agarose in a glass-bottom cell tissue culture dish (Fluorodish, World Precision Instruments, USA). An inverted confocal microscope Olympus FV-1000 was used for imaging, employing a 20 x objective (NA 0.75) or a 40 x oil immersion objective (NA 1.30). Z-volumes were acquired with a 0.5 µm resolution, and images were processed and analyzed using ImageJ and Imaris. The distances between the DFCs and the EVL at 6 and 8 hpf were measured in fixed whole-mount samples. Using ImageJ, the distances were analyzed from the center of the leading DFCs to the leading edge of the EVL, as marked by ZO-1. The area of ZO-1 enrichments was measured using Fiji in cropped images that included only DFCs and at 6 hpf also the apical part of the EVL.

## Fluorescent microspheres injection and analysis

FluoSpheresH fluorescent microspheres injection was conducted as previously described (*Borovina et al., 2010*). Embryos were embedded in 1% low melting-point agarose diluted in egg water. Imaging was performed with a 40 x water dipping objective using an upright Yokogawa CSU-X1 spinning disk scan head, mounted on a DM6000 upright Leica microscope and a CCD CoolSnap HQ2 camera at 4 Hz using the green channel at 0.25 µm resolution on a single Z-plane. Beads trajectory tracking was performed using an already developed toolbox (*Sbalzarini and Koumoutsakos, 2005*). The size detection parameter was set to 0.75–1.25 µm objects with a brightness superior to the 0.01 percentile of the whole image. Objects that traveled less than 2.5 µm between two consecutive frames were linked into the same trajectory and considered as a single bead. Objects traveling longer distances were analyzed as separate trajectories, even if they originated from the same bead. Analyses were then performed using a custom-made Matlab code. For better visualization, trajectory plots only show 20 tracked beads. MSD analysis was conducted using an already published code (https://tinevez. github.io/msdanalyzer/) (*Tarantino et al., 2014*). Average MSD plots only show particles tracked between 1 and 8 s. Very short tracking durations do not permit correctly estimating displacement properties, and less than 1% of the particles were tracked for more than 8 s.

## Data analysis

For analysis and quantification, mutants, WT controls, and mutants were coded prior to quantification and decoded after for analysis. Expression analysis, DFC clustering, and heart and visceral organ positioning experiments were conducted at least three independent times, each with a comparable N number, to ensure reproducibility and account for batch effects. The reported data represent pooled results from these independent experiments. Experiments for heart looping, *dand5* expression, and visceral organ phenotypes have multinomial outcomes, and the *p*-values for these distributions were evaluated accordingly. For any given pair of conditions α (control) and β (mutated condition), the number of observed experimental outcomes was counted. The number of outcomes for each condition was denoted as (nα, dα, rα) and (nβ, dβ, rβ), with Nα and Nβ denoting the total number of observations. The frequency of each outcome under a particular condition is simply the number of

observations for the given outcome divided by the total number of observations. The dissimilarity between two conditions α and β was measured as the sum of absolute differences between their respective frequencies δ = |fnα - fnβ| + |fdα - fdβ| + |frα - frβ|. Then the probability of the null hypothesis was estimated, i.e., that the multinomial distributions of outcomes related to the two conditions are the same, as the probability of obtaining a dissimilarity equal to or higher than the one observed by simply pooling all the observations together, and again randomly subdivided in two sets of observations of size Nα and Nβ. To numerically estimate this probability, the procedure of pooling the observations together was repeated for K=100,000 times, randomly separating them again in two sets, and evaluating the new dissimilarity δ'. The probability of the null hypothesis is then given by the numerical frequency with which the dissimilarity δ' ≥ δ is greater than or equal to the one observed in the experiments. (*Zwillinger, 2022*; *Papoulis, 1984*).

Source data as well as Python and Matlab codes used for beads and DFC tracking are available at https://doi.org/10.5281/zenodo.15622175. Codes for statistical analysis are available at https://github.com/FannyEggeler/Meteorin_analysis (copy archived at *Eggeler, 2024*).

## Phylogenetic analysis

Protein sequences were obtained from the Ensembl Genome Browser or UniProt. The following sequences were used: *D. rerio* Metrn (ENSDARP00000041804.5), *D. rerio* Metrnla (ENSDARP00000131064.1), *D. rerio* Metrnlb (ENSDARP00000126894.1), *G. gallus* Metrn (ENSGALP00010026416.1), *G. gallus* Metrnl (ENSGALP00010044844.1), *H. sapiens* Metrn (ENSP00000455068.1), *H. sapiens* Metrnl (ENSP00000315731.6), *M. musculus* Metrn (ENSMUSP00000127275.2), *M. musculus* Metrnl (ENSMUSP00000038126.8), *X. leavis* Metrnl (ENSXETP00000107425.1), *L. oculatus* Metrn (ENSLOCP00000003703.1), *L. oculatus* Metrnl (ENSLOCP00000017224.1), *B. lanceolatum* (A0A8K0F3K3), *S. clava* (XP_039261436). *L. oculatus* was chosen as an outgroup, *B. lanceolatum* as an example for Cephalochordata and *S. clava* as a representative for Tunicata. The phylogenetic tree was reconstructed with the phylogeny analysis from https://www.phylogeny.fr/ (*Dereeper et al., 2008*; *Dereeper et al., 2010*; *Edgar, 2004*; *Castresana, 2000*; *Guindon and Gascuel, 2003*; *Anisimova and Gascuel, 2006*; *Chevenet et al., 2006*).

## Resource availability

### Lead contact

Requests for further information and resources should be directed to and will be fulfilled by the lead contact, Filippo Del Bene (filippo.del-bene@inserm.fr).

### Materials availability

All unique/stable reagents generated in this study are available from the lead contact with a completed materials transfer agreement.

### Data and code availability

- All data generated or analysed during this study are included in the manuscript and supporting files; source data files have been provided for *Figure 1—figure supplement 1*, *Figure 2* and *Figure 6—figure supplement 2*.
- This paper does report original code which is available at https://doi.org/10.5281/zenodo.15622175. Codes for statistical analysis are available at https://github.com/FannyEggeler/Meteorin_analysis (copy archived at *Eggeler, 2024*).
- Any additional information required to reanalyze the data reported in this paper is available from the lead contact upon request.

## Acknowledgements

We thank all members of the Del Bene lab for discussions, the Institut Curie and Institut de la Vision imaging and animal facilities. We acknowledge the lab of Jean Livet for their support with the chick analysis. We would like to thank Pierre Luc Bardet from IBPS (Sorbonne Université) for fruitful discussions. We thank Clémence Gentner from Gilles Tessier's lab at the Institut de la Vision for the microspheres sample. We thank Marco Molari (Biozentrum Basel) for the support with the statistical

analysis. Work in the Del Bene laboratory was supported by ANR MetAxon [ANR-17-CE16-0007], the IHU FOReSIGHT [ANR-18-IAHU-0001], CNRS, INSERM, and Sorbonne Université core funding. FE was supported by the École des Neurosciences de Paris (ENP) Ile-de-France network and the Fondation pour la Recherche Médicale (FRM). FDS was supported by a doctoral fellowship of the Curie International PhD program. TOA was supported by a Boehringer Ingelheim Fonds PhD Fellowship. *Figures 4C and 6B* were created in BioRender. Albadri, S (2025) https://BioRender.com/al0okfi".

## Additional information

### Competing interests

Flavia De Santis: is affiliated with ZeClinics SL. The author has no other competing interests to declare. Filippo Del Bene: Reviewing editor, eLife. The other authors declare that no competing interests exist.

### Funding

| Funder | Grant reference number | Author |
|---|---|---|
| Agence Nationale de la Recherche | MetAxon [ANR-17-CE16-0007] | Filippo Del Bene |
| Agence Nationale de la Recherche | IHU FOReSIGHT [ANR-18-IAHU-0001] | Filippo Del Bene |

The funders had no role in study design, data collection and interpretation, or the decision to submit the work for publication.

### Author contributions

Fanny Eggeler, Conceptualization, Resources, Data curation, Formal analysis, Investigation, Visualization, Methodology, Writing – original draft, Writing – review and editing; Jonathan Boulanger-Weill, Conceptualization, Software, Formal analysis, Validation, Investigation, Visualization, Methodology; Flavia De Santis, Conceptualization, Resources, Formal analysis, Investigation, Visualization, Methodology, Writing – review and editing; Laura Belleri, Data curation, Formal analysis, Investigation, Methodology, Writing – review and editing; Karine Duroure, Data curation, Investigation, Methodology, Writing – review and editing; Thomas O Auer, Conceptualization, Resources, Methodology, Writing – review and editing; Shahad Albadri, Conceptualization, Supervision, Validation, Investigation, Visualization, Methodology, Writing – original draft, Writing – review and editing; Filippo Del Bene, Conceptualization, Supervision, Funding acquisition, Validation, Writing – original draft, Project administration, Writing – review and editing

### Author ORCIDs

Fanny Eggeler ⬤ http://orcid.org/0009-0001-6803-1984
Jonathan Boulanger-Weill ⬤ http://orcid.org/0000-0002-1580-0778
Flavia De Santis ⬤ http://orcid.org/0000-0002-0988-6560
Laura Belleri ⬤ http://orcid.org/0009-0001-5475-7821
Karine Duroure ⬤ http://orcid.org/0000-0002-5959-569X
Thomas O Auer ⬤ http://orcid.org/0000-0003-3442-0567
Shahad Albadri ⬤ https://orcid.org/0000-0002-3243-7018
Filippo Del Bene ⬤ https://orcid.org/0000-0001-8551-2846

### Ethics

All animal procedures were performed in accordance with French and European Union animal welfare guidelines with protocols approved by the committee on ethics of animal experimentation of Sorbonne Université (APAFIS#21323- 2019062416186982).

Reviewer #1 (Public review): https://doi.org/10.7554/eLife.105430.3.sa1
Reviewer #1 (Public review): https://doi.org/10.7554/eLife.105430.3.sa2
Author response https://doi.org/10.7554/eLife.105430.3.sa3

## Additional files

**Supplementary files**
MDAR checklist

**Data availability**
All data generated or analysed during this study are included in the manuscript and supporting files; source data files have been provided for *Figure 1—figure supplement 1*, *Figure 2* and *Figure 6—figure supplement 2*.

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
