## [Editor Report · eLife Assessment]

This study presents **important** insights into the regulation of left-right organ formation. By combining genetic perturbation of all three Meteorin genes in zebrafish and timelapse imaging, the authors identify an essential role for this protein family in the establishment of left-right patterning. They provide **convincing** evidence that Meteorins are required for the morphogenesis of dorsal forerunner cells, the precursors of the left-right organizer (also named Kupffer's vesicle) in zebrafish. In line with this, Meteorins were shown to genetically interact with integrins ItgaV and Itgb1b to regulate dorsal forerunner cell clustering.

---

## [Referee Report · Reviewer #1 (Public review)]

Summary:

Meteorin proteins were initially described as secreted neurotrophic factors. In this manuscript, Eggeler et al. demonstrate a novel role for Meteorins in establish left-right axis formation in the zebrafish embryo. The authors generated null mutations in each of the three zebrafish meteorin genes - metrn, metrnla, and metrnlab. Triple mutant embryos displayed phenotypes strongly associated with left-right defects such as heart looping and visceral organ placement, and disrupted expression of Nodal-responsive genes, as did single mutants for metrn and metrnla. The authors then go on to demonstrate that these defects in left-right asymmetry are likely to due to defects in Kupffer's Vesicle and the progenitor dorseal forerunner cells including impaired lumen formation and reduced fluid flow, reduced clustering among DFCs, impaired DFC migration, mislocalization of apical proteins ZO-1 and aPKC, and detachment of DFCs from the EVL. Notably, the authors found that expression of marker genes sox32 and sox17 were not affected, suggesting Meteorins are required for DFC/KV morphogenesis but not necessarily fate specification. Finally, the authors show genetic interaction between Meteorins and integrin receptors, which were previously implicated in left-right patterning. In a supplemental figure, the manuscript also presents data showing expression of meteorin genes around the chick Hensen's node, suggesting that the left-right patterning functions may be conserved among vertebrates.

Strengths:

Strengths of this study include the generation of a triple mutant line that targets all known zebrafish meteorin family members. The experiments presented in this study were rigorous especially with respect to quantification and statistical analysis.

Weaknesses:

Although the authors convincingly demonstrate a role for Meteorins in zebrafish left-right patterning, data supporting a conserved role in other vertebrates is compelling but limited to one supplemental figure. This aspect would be interesting to follow up in future studies.

Comments on revisions:

I thank the authors for their thoughtful responses to the reviewers. They have adequately addressed all of my concerns.

---

## [Referee Report · Reviewer #1 (Public review)]

Summary:

Meteorin proteins were initially described as secreted neurotrophic factors. In this manuscript, Eggeler et al. demonstrate a novel role for Meteorins in establish left-right axis formation in the zebrafish embryo. The authors generated null mutations in each of the three zebrafish meteorin genes - metrn, metrnla, and metrnlab. Triple mutant embryos displayed phenotypes strongly associated with left-right defects such as heart looping and visceral organ placement, and disrupted expression of Nodal-responsive genes, as did single mutants for metrn and metrnla. The authors then go on to demonstrate that these defects in left-right asymmetry are likely to due to defects in Kupffer's Vesicle and the progenitor dorseal forerunner cells including impaired lumen formation and reduced fluid flow, reduced clustering among DFCs, impaired DFC migration, mislocalization of apical proteins ZO-1 and aPKC, and detachment of DFCs from the EVL. Notably, the authors found that expression of marker genes sox32 and sox17 were not affected, suggesting Meteorins are required for DFC/KV morphogenesis but not necessarily fate specification. Finally, the authors show genetic interaction between Meteorins and integrin receptors, which were previously implicated in left-right patterning. In a supplemental figure, the manuscript also presents data showing expression of meteorin genes around the chick Hensen's node, suggesting that the left-right patterning functions may be conserved among vertebrates.

Strengths:

Strengths of this study include the generation of a triple mutant line that targets all known zebrafish meteorin family members. The experiments presented in this study were rigorous especially with respect to quantification and statistical analysis.

Weaknesses:

Although the authors convincingly demonstrate a role for Meteorins in zebrafish left-right patterning, data supporting a conserved role in other vertebrates is compelling but limited to one supplemental figure. This aspect would be interesting to follow up in future studies.

Comments on revisions:

I thank the authors for their thoughtful responses to the reviewers. They have adequately addressed all of my concerns.

---

## [Author Response]

The following is the authors’ response to the original reviews

**Public Review:**

**Reviewer #1 (Public review):**
Summary:Meteorin proteins were initially described as secreted neurotrophic factors. In this manuscript, Eggeler et al. demonstrate a novel role for Meteorins in establish left-right axis formation in the zebrafish embryo. The authors generated null mutations in each of the three zebrafish meteorin genes - metrn, metrnla, and metrnlab. Triple mutant embryos displayed phenotypes strongly associated with left-right defects such as heart looping and visceral organ placement, and disrupted expression of Nodal-responsive genes, as did single mutants for metrn and metrnla. The authors then go on to demonstrate that these defects in left-right asymmetry are likely to due to defects in Kupffer's Vesicle and the progenitor dorseal forerunner cells including impaired lumen formation and reduced fluid flow, reduced clustering among DFCs, impaired DFC migration, mislocalization of apical proteins ZO-1 and aPKC, and detachment of DFCs from the EVL. Notably, the authors found that expression of marker genes sox32 and sox17 were not affected, suggesting Meteorins are required for DFC/KV morphogenesis but not necessarily fate specification. Finally, the authors show genetic interaction between Meteorins and integrin receptors, which were previously implicated in left-right patterning. In a supplemental figure, the manuscript also presents data showing expression of meteorin genes around the chick Hensen's node, suggesting that the left-right patterning functions may be conserved among vertebrates.Strengths:Strengths of this study include the generation of a triple mutant line that targets all known zebrafish meteorin family members. The experiments presented in this study were rigorous, especially with respect to quantification and statistical analysis.Weaknesses:Although the authors convincingly demonstrate a role for Meteorins in zebrafish left-right patterning, data supporting a conserved role in other vertebrates is compelling but limited to one supplemental figure.

We thank the reviewer for their thoughtful summary of our study and for highlighting the strengths of our work, including the generation of the triple mutant line and the rigor of our experimental design and quantitative analyses. We also appreciate the constructive feedback regarding the limited functional data supporting the conservation of Meteorin function in other vertebrates. We agree that this is an important aspect that could be further explored. While functional studies in additional species are beyond the current scope, we will consider such experiments in future work.

We would like to highlight the phylogenetic analysis of Meteorin proteins we have already performed and included in the manuscript (Fig. S7D), which illustrates the evolutionary conservation of this protein family and supports the possibility of a conserved role in left-right patterning.

Additionally, we have expanded the methods and discussion to include: (1) details on zebrafish viability in contrast to reported embryonic lethality in *metrn* mutant mice, (2) the background strains used in our study, (3) observed variability in DFC number and potential batch effects and (4) clarification of our 'convergence ratio' quantification approach.

**Reviewer #2 (Public review):**
Summary:In this manuscript the authors describe their study on the role of meteorins in establishing the left-right organizer. The left-right organizer is a transient organ in vertebrate embryos in which rotating cilia cause a fluid flow that breaks the left-right symmetry and coordinates lateralization of internal organs such as gut and heart. In zebrafish, the left-right organizer (also named Kupffer's vesicle) is formed by dorsal forerunner cells, but very little is known about how dorsal forerunner cells coalles and form this ciliated vesicle in the embryo. The authors mutated the three meteorin-coding genes in zebrafish and observed that mutations in each one of these causes laterality defects with the strongest defects observed in the triple mutant. Loss of meteorins affects nodal gene expression, which play essential roles in establishing organ laterality. Meteorins are widely expressed in developing embryos and expression in lateral plate mesoderm and dorsal forerunner cells was observed. The meteorin triple mutant embryos display defects in the migration and clustering of the dorsal forerunner cells impairing kupffer's vesicle formation and cilia rotation. Finally, the authors show that meteorins genetically interact with integrins.Strengths:- These authors went through the lengthy process of generating triple mutants affecting all three meteorin genes. This provides robust genetic evidence on the role of meteorins in establishing organ laterality and circumvented that interpretation of the results would be hard due to redundant functions of meteorins.- The use of life imaging on triple mutants is appreciated- High-quality imaging of dorsal forerunner to quantify cell migrations and its relation to Kupffer's vesicle formation.Weaknesses:- Lack of a model how meteorins regulate dorsal forerunner cell migration.- Only genetic data to suggest a link between meteorins and integrins- Besides its role in DFC migration, meteorins may also play a more direct role in regulating Nodal signaling, which is not addressed here.

We appreciate the recognition of the strengths of our study, particularly the generation of the triple meteorin mutants and the use of high-resolution imaging to quantify DFC behavior and Kupffer’s vesicle formation—both of which were central to providing robust evidence for Meteorins' role in left-right patterning.

We also value the reviewer’s comments on areas that need further exploration, including the need for a mechanistic model explaining how Meteorins regulate DFC migration, the genetic interaction with integrins, and the potential direct involvement of Meteorins in Nodal signaling.

We agree that deeper mechanistic insights would strengthen the study. While our findings suggest that Meteorins influence DFC migration and clustering through integrin pathways, a detailed mechanistic dissection, particularly regarding the yet unidentified Meteorin receptor, lies beyond the current scope. However, we consider this a key aspect for future research and have discussed it further in the revised discussion section.

In response to the reviewer’s suggestions, we have expanded the discussion to address the limitations of the current data linking Meteorins and integrins, including relevant citations to studies that implicate integrins in similar contexts. Additionally, we have added a more detailed discussion of the potential for Meteorins to directly influence Nodal signaling, and we cite a relevant study to support this possibility.

Once again, we thank the reviewer for their insightful and constructive comments. These points raise important directions for future investigation that will further advance our understanding of Meteorin function in left-right axis formation.

**Recommendations for the authors:**

**Reviewer #1 (Recommendations for the authors):**
In the Results section (p. 9), the authors state, "...a reduced ZO-1 enrichment at the apical junctions of triplMUT GFP-positive DFCs could be detected." However, in Fig. 4F-G, the areas of ZO-1 enrichment indicated by arrowheads appear quite far from the DFCs themselves, making it unclear if these ZO-1-enriched areas are apical DFC junctions (as stated in the text) or instead are part of the EVL. Is it possible to include an additional cell membrane marker or other landmarks? In addition, the differences in ZO-1 accumulation between mutants and WT appear relatively modest. Is it possible to provide quantification of this effect?

We appreciate the reviewer’s request for additional stainings and further clarification and we would like to highlight the requested quantifications of ZO-1 accumulation, including statistical analysis, are already provided in Fig. S5E.

In mouse, loss of Meteorin is embryonic lethal yet the zebrafish triple mutants are viable. Could the authors discuss this discrepancy?

We have expanded the discussion to address this point, suggesting that species-specific differences in compensatory mechanisms may explain the observed differences in viability. We would like to reiterate that while one study has reported embryonic lethality in metrn mutant mice, this specific mouse line has not been further investigated in any recent publications. Additionally, in collaboration with the lab of Alain Chédotal, we generated independent *metrn* and *metrnl* mutant mouse lines, which did not exhibit the phenotype described in the previously mentioned study.

It has been reported that TL and AB strains exhibit variable numbers of DFCs and thus laterality defects (Moreno-Ayala et al., 2021, Cell Reports 34(2):108606). Would it be possible for the authors to report background stains used in this study and those used to generate the meteorin knock-outs?

We appreciate the comment highlighting the importance of specifying the background strains used in our study. We have now included this information in the methods section, detailing the zebrafish strains utilized throughout our experiments.

For statistical analysis, would be possible for the authors to report the number of clutches examined to control for batch effects (especially given the wide variability in DFC numbers as noted above)?

For further clarification, we have now included additional explanation on number of clutches in the methods section.

In the Methods section (p. 19), the description of how the convergence ratio was computed was somewhat unclear. Could the authors provide a citation or include a diagram/schematic?

We have revised the Methods section to provide a clearer definition of the convergence ratio and have included a schematic (Fig. 4D) to illustrate how it was calculated.

**Reviewer #2 (Recommendations for the authors):**
- Meteorins are widely expressed in the embryo. Can the authors comment on whether meteorin expression is required in the dorsal forerunner cells (DFCs) or in other cells? This could be addressed by knockdown experiments in DFCs as described by others (PMID: 15716348)

We thank the reviewer for this important comment. In our study, we have shown that Meteorins are not required for the identity of DFCs, as several DFC-specific markers remain expressed in the respective cells within the meteorin mutant background (see Fig. S4).

- In fig1d and 1e the authors use heterotaxy to describe visceral organ placement. The embryo shown in 1d seems to display situs inversus instead of heterotaxy, which is defined as discordance in organ position. The authors should clarify this.

We agree with the reviewer and have revised the figures and figure legends to clarify the distinction between *situs inversus* and heterotaxy.

- In Fig2 the authors show that nodal pathway genes are reduced, suggesting reduced Nodal signaling. How do they explain this as loss of cilia rotation generally leads to randomization of Nodal signaling but not a reduction in signaling.

Following this suggestion we have now added a further discussion on the possibility that Meteorins could directly regulate Nodal signaling in addition to their role in DFC migration and have cited a relevant study.

- Reduced Nodal signaling in the LPM leads to organ laterality defects. Most anterior tissues like the heart are more sensitive to perturbation in Nodal signaling in the LPM compared to more posterior organs like gut (see also PMID: 25684355). Since in triple mutants the position of the heart is more affected than the position of the visceral organs this suggests that meteorins play an additional role in Nodal signaling in the LPM. As others have shown that meteorins regulate nodal activity (PMID: 24558432), the authors should address this further.

As described above, we have now added a further discussion on the possibility that Meteorins could directly regulate Nodal signaling in addition to their role in DFC migration and have cited a relevant study. Further investigation into a possible direct role of Meteorins in Nodal signaling will be pursued in future work.

- The term 'convergence ratio' is not clearly described and confusing as convergence is also used for the movement of LPM cells towards the midline.

As noted in response to Reviewer #1, we have revised the Methods section and included a schematic in Fig. 4D to better explain this parameter.

We are grateful for the thoughtful critiques from both reviewers, which have been very constructive and improved the clarity of our study. We believe that the revisions we have made address the concerns raised, and we look forward to your evaluation of our revised manuscript.